# Privacy Amplification Through Synthetic Data: Insights from Linear Regression

**Clément Pierquin** [1 2]  **Aurélien Bellet** [3]  **Marc Tommasi** [2]  **Matthieu Boussard** [1]

## Abstract

Synthetic data inherits the differential privacy guarantees of the model used to generate it. Additionally, synthetic data may benefit from *privacy amplification* when the generative model is kept hidden. While empirical studies suggest this phenomenon, a rigorous theoretical understanding is still lacking. In this paper, we investigate this question through the well-understood framework of linear regression. First, we establish negative results showing that if an adversary controls the seed of the generative model, a single synthetic data point can leak as much information as releasing the model itself. Conversely, we show that when synthetic data is generated from random inputs, releasing a limited number of synthetic data points amplifies privacy beyond the model's inherent guarantees. We believe our findings in linear regression can serve as a foundation for deriving more general bounds in the future.

## 1. Introduction

Differential privacy (DP) (Dwork and Roth, 2014) has become the gold standard for privacy-preserving data analysis. Training machine learning models with DP guarantees can be achieved through various techniques: *output perturbation* (Chaudhuri et al., 2011; Zhang et al., 2017b; Lowy and Razaviyayn, 2024), which adds noise to the non-private model; *objective perturbation* (Chaudhuri et al., 2011; Kifer et al., 2012; Redberg et al., 2023), which introduces noise into the objective function; and *gradient perturbation*, which injects noise into the optimization process, as in DP-SGD (Song et al., 2013; Bassily et al., 2014; Abadi et al., 2016; Feldman et al., 2018). Once trained, the model can be safely released, with its privacy guarantees extending to all subsequent uses thanks to the post-processing property of DP. This is partic-

ularly relevant for *differentially private generative models* (Zhang et al., 2017a; Xie et al., 2018; McKenna et al., 2019; Jordon et al., 2019; McKenna et al., 2021; Lee et al., 2022; Dockhorn et al., 2023; Bie et al., 2023), where the synthetic data they produce inherits the same privacy guarantees as the model itself.

Empirical studies, however, suggest that synthetic data may offer even stronger privacy protection than the theoretical guarantees provided by the model (Annamalai et al., 2024). This suggests that certain structural properties of the data or the generative process itself could contribute to an implicit privacy amplification effect. One possible intuition is that the privacy leakage might be reduced when the number of released synthetic data points is "small" relative to the complexity of the generative model. However, to the best of our knowledge, no existing work has formally established the existence of such a privacy amplification effect, and a rigorous quantification of differential privacy in synthetic data release remains an open question.

To address this gap, this paper takes an initial step towards developing a theoretical framework for quantifying privacy in synthetic data release. We focus on the well-studied setting of (high-dimensional) linear regression trained via a least-squares objective as a simple case study. This model has the advantage of being analytically tractable but sufficiently expressive to capture phenomena observed in more complex models—such as double descent in overparameterized regimes (Hastie et al., 2022) and, more recently, model collapse in generative AI (Dohmatob et al., 2024; Gerstgrasser et al., 2024).

We rely on the $f$-Differential Privacy ($f$-DP) framework (Dong et al., 2022), which provides a flexible and robust approach to privacy analysis, allowing precise characterizations of privacy guarantees through trade-off functions. When these trade-offs functions are difficult to interpret, we also express privacy guarantees in the Rényi differential privacy (RDP) framework (Mironov, 2017).

Our results are two-fold. First, in Section 3, we present negative results in scenarios where an adversary controls the seed of the synthetic data generation process. Specifically, we show that the adversary can leverage this control to achieve privacy leakage equivalent to the bound imposed by post-processing the model using only a single synthetic

[1]Craft AI, Paris, France [2]Université de Lille, Inria, CNRS, Centrale Lille, UMR 9189 CRIStAL, F-59000 Lille, France [3] Inria, Université de Montpellier, INSERM. Correspondence to: Clément Pierquin <clement.pierquin@craft-ai.fr>.

*Proceedings of the 42nd International Conference on Machine Learning*, Vancouver, Canada. PMLR 267, 2025. Copyright 2025 by the author(s).

sample. Second, in Section 4, we analyze the privacy guarantees when synthetic data is generated from random inputs to a private regression model obtained via output perturbation. We demonstrate that privacy amplification is possible in this setting, depending on the model size and the number of released synthetic samples. All proofs can be found in the supplementary material.

Our findings highlight the critical role of the randomness given as input to the model, which must remain concealed from the adversary in order to enable privacy amplification. While the practical impact of our results is limited, we believe they offer valuable insights and lay the groundwork for a deeper understanding of synthetic data privacy in more complex machine learning models.

**Related work.** To the best of our knowledge, existing methods for differentially private synthetic data generation rely on learning a differentially private generative model (Hu et al., 2024). Early approaches focused on marginal-based techniques for tabular data, where a graphical model—such as a Bayesian network—is privately estimated from data and then used to generate new samples (Zhang et al., 2017a; McKenna et al., 2019; 2021). More recent methods extend to other data modalities, leveraging expressive neural network-based generative models—like GANs and diffusion models—trained with differentially privacy (Xie et al., 2018; Jordon et al., 2019; Lee et al., 2022; Dockhorn et al., 2023; Bie et al., 2023). A key advantage of neural networks is the availability of general differentially private training algorithms, such as DP-SGD (Abadi et al., 2016) and PATE (Papernot et al., 2017), which can be applied across various generative models. Crucially, all these methods rely on the post-processing theorem to ensure the privacy guarantees of the generated synthetic data—but it remains unclear whether this guarantee is tight or potentially overly conservative.

In principle, one could deviate from this dominant approach by adding noise directly to the data generated by a (non-private) generative model. In such cases, the overall privacy loss would scale with the number of released data points due to the composition property of differential privacy. However, this approach would require strong and often unrealistic assumptions about the data. Most critically, it would lead to significant utility loss—particularly for high-dimensional perceptual data such as images, where even small perturbations can severely degrade semantic content and downstream performance. To our knowledge, no successful applications of this approach have been demonstrated in practice.

Interestingly, our results in Section 4 suggest that differentially private generative models may offer the best of both worlds: the post-processing guarantee, which strictly bounds the privacy leakage when releasing a large number of samples, *and simultaneously* a privacy guarantee that scales with the number of released data points, which is more favorable when only a few samples are released.

Our results relate to the concept of *privacy amplification* (Balle et al., 2018; Feldman et al., 2018; Erlingsson et al., 2019; Cyffers and Bellet, 2022), which leverages the non-disclosure of certain intermediate computations to strengthen the privacy guarantees of existing mechanisms. We note that the form of amplification we study in the context of synthetic data release differs from privacy amplification by iteration (Feldman et al., 2018). In that setting, the final model is released after private training. In contrast, our approach withholds the model entirely and releases only synthetic data generated from random inputs to the model, introducing an additional layer of privacy protection.

We conclude our discussion of related work by mentioning a recent study that shows synthetic data can satisfy differential privacy guarantees without formal guarantees for the generative model itself (Neunhoeffer et al., 2024). However, this work is limited to a simple model where the private training data is one-dimensional, and the synthetic data is generated from a Gaussian distribution with mean and variance estimated from the private data. In contrast, our paper addresses a different, more complex problem: we investigate the privacy guarantees associated with releasing the output of a differentially private model, specifically linear regression. In our case, we directly model the distribution of the output of linear regression for a random seed, which corresponds to a product of Gaussian matrices.

## 2. Background & Preliminaries

### 2.1. Differential Privacy

In this section, we give a brief background about the technical tools we use from the differential privacy literature. Here and throughout, $\mathcal{M}$ denotes a randomized mechanism, and we say that two datasets $\mathcal{D}$ and $\mathcal{D}'$ of fixed size $m$ are adjacent if they differ in a single data point.

**Rényi Differential Privacy (RDP)** is a variant of DP which quantities the privacy guarantees in terms of a Rényi divergence between $\mathcal{M}(\mathcal{D})$ and $\mathcal{M}(\mathcal{D}')$ (Mironov, 2017). Formally, for $\alpha > 1$, let $D_\alpha(P, Q) = \frac{1}{\alpha-1} \log \mathbb{E}_{x \sim P}\left[\left(\frac{Q(x)}{P(x)}\right)^\alpha\right]$ be the Rényi divergence of order $\alpha$ between distributions $P$ and $Q$. By abuse of notation, if $V \sim P$, $W \sim Q$, we will write $D_\alpha(V, W) = D_\alpha(P, Q)$. For $\varepsilon > 0$, a mechanism $\mathcal{M}$ is said to satisfy $(\alpha, \varepsilon)$-RDP if, for any two adjacent datasets $\mathcal{D}$ and $\mathcal{D}'$, we have:

$$D_\alpha(\mathcal{M}(\mathcal{D})\|\mathcal{M}(\mathcal{D}')) \leq \varepsilon.$$

**Trade-off functions and $f$-DP.** Trade-off functions (Dong et al., 2022) capture the inherent trade-off between type I and type II errors of hypothesis tests that distinguish between outputs generated from two adjacent datasets. Let $P$ and $Q$

be two distributions and consider the following hypothesis testing problem:

$$H_1 : \text{the distribution is } P \quad \text{vs} \quad H_2 : \text{the distribution is } Q.$$

For a given rejection rule $\phi \in [0,1]$, the type I error is $\mathbb{E}_P[\phi]$ and the type II error is $1 - \mathbb{E}_Q[\phi]$. Then, the trade-off function $T(P,Q)$ is defined as:

$$T(P,Q)(\alpha) = \inf_{\phi}\{1 - \mathbb{E}_Q[\phi] : \mathbb{E}_P[\phi] \le \alpha\}.$$

Again, by abuse of notation, if $V \sim P$, $W \sim Q$, we will write $T(V,W) = T(P,Q)$.

Let $f : [0,1] \to [0,1]$ be a decreasing convex function. A mechanism $\mathcal{M}$ is said to satisfy $f$-Differential Privacy ($f$-DP) if for any adjacent $\mathcal{D}, \mathcal{D}'$, we have:

$$T(\mathcal{M}(\mathcal{D}), \mathcal{M}(\mathcal{D}')) \ge f.$$

Gaussian Differential Privacy (GDP) (Dong et al., 2022) is a special case of $f$-DP defined with Gaussian trade-off functions. For $\mu \ge 0$, we define:

$$G_\mu = T(\mathcal{N}(0,1), \mathcal{N}(\mu,1)).$$

Then, $\mathcal{M}$ is said $\mu$-GDP if it is $G_\mu$-DP.

### 2.2. Linear Regression Setting

Throughout the paper, we consider a multi-output linear regression setting, where a dataset $\mathcal{D} = (X,Y) = ((x_1,y_1), \ldots, (x_m,y_m))$ consists of $m$ labeled data points, with $X \in \mathcal{X}^{d \times m}$ the data matrix and $Y \in \mathbb{R}^{n \times m}$ the label (output) matrix. Here, $d$ denotes the number of features of the input (private) data points, and $n$ denotes the dimension of outputs (i.e., the number of features of synthetic data points). A linear model is represented by parameters $w \in \mathbb{R}^{n \times d}$ and predicts $\hat{Y} = wX$.

## 3. Releasing Synthetic Data from Fixed Inputs

In this section, we investigate the privacy guarantees of releasing synthetic data when the input to the generation process is fixed and chosen by an adversary. This represents a strong yet practically relevant threat model, encompassing situations such as when an adversary has access to an API that allows them to query the generative model.

Focusing on linear regression, we consider two standard ways to train the model with differential privacy guarantees: output perturbation (Chaudhuri et al., 2011) and Noisy Gradient Descent (Song et al., 2013; Bassily et al., 2014; Abadi et al., 2016). In both cases, we show that when the adversary controls the seed of the generation process, they can induce privacy leakage that reaches the upper bound

established by post-processing, even with just a single data point. In other words, no privacy amplification occurs under this scenario. These negative results underscore the vulnerability of synthetic data generation mechanisms to adversarial manipulation.

### 3.1. Output Perturbation

We begin with models trained with output perturbation. We make the assumption that each element $(x_i, y_i)$ of the dataset satisfies the following property: $\|x_i\| \le M_x$ and $\|y_i\| \le M_y$, where $\|\cdot\|$ denotes the Frobenius norm. These hypotheses are standard in private linear regression (Wang, 2018). Denoting the $\ell_2$-regularized least-square objective as $F_\lambda(w; \mathcal{D}) = \frac{1}{m}\sum_{k=1}^{m}\|wx_i - y_i\|^2 + \lambda\|w\|^2$, we know from (Chaudhuri et al., 2011) that $\mathcal{D} \mapsto \arg\min_{w \in \mathbb{R}^{n \times d}} F_\lambda(w; \mathcal{D})$ has bounded sensitivity $\Delta = 2L/m\lambda$, where $L = M_x^2 M_\theta + M_x M_y + \lambda M_\theta$ and $M_\theta$ is the upper bound of the Frobenius norm of the minimizer of $F_\lambda$, always bounded for ridge regression. We denote the output perturbation mechanism by $\mathcal{M}(\mathcal{D}) = \arg\min_{w \in \mathbb{R}^{n \times d}} F_\lambda(w; \mathcal{D}) + \sigma_\theta N$, where $N_{ij} \stackrel{iid}{\sim} \mathcal{N}(0,1)$.

For two adjacent datasets $\mathcal{D}$ and $\mathcal{D}'$, let $v^* = \arg\min_{w \in \mathbb{R}^{n \times d}} F_\lambda(w; \mathcal{D})$ and $w^* = \arg\min_{w \in \mathbb{R}^{n \times d}} F_\lambda(w; \mathcal{D}')$. With some abuse of notation, we denote $\mathcal{M}(v^*) = \mathcal{M}(\mathcal{D})$ and $\mathcal{M}(w^*) = \mathcal{M}(\mathcal{D}')$.

The exact trade-off function of output perturbation with the Gausian mechanism is well known (Dong et al., 2022):

$$\inf_{\substack{v,w \in \mathbb{R}^{n \times d}: \\ \|v-w\| \le \Delta}} T(\mathcal{M}(v), \mathcal{M}(w)) = \inf_{\substack{\mu \in \mathbb{R}^{n \times d}: \\ \|\mu\| \le \Delta}} G_{\|\mu\|/\sigma_\theta} = G_{\Delta/\sigma_\theta}.$$

The RDP guarantees are also known (Mironov, 2017):

$$\sup_{\substack{v,w \in \mathbb{R}^{n \times d}: \\ \|v-w\| \le \Delta}} D_\alpha(\mathcal{M}(v), \mathcal{M}(w)) = \frac{\alpha\Delta^2}{2\sigma_\theta^2}.$$

Due to translation invariance of trade-off function for Gaussian matrices and noting $\mu = w^* - v^*$, comparing $\mathcal{M}(v^*)$ and $\mathcal{M}(w^*)$ is equivalent to comparing $V = \mathcal{M}(0_{n \times d})$ and $W = V + \mu$ where $\mu$ is called the shift between $V$ and $W$.

We are interested in quantifying the privacy leakage of releasing the output of the model queried with a seed input. Formally, a seed is a vector $z \in \mathbb{R}^d$ and we define the corresponding query to model $v^*$ (respectively $w^*$) as $v^*z \in \mathbb{R}^n$ (respectively $w^*z$). Note that $\mathcal{M}(v^*)z$ is a Gaussian vector. Based on the derivation above and due to translation invariance of trade-off functions for Gaussian vectors, quantifying the privacy leakage then amounts to characterizing the trade-off function $T(Vz, Wz)$.

It is clear that an adversary can recover the model parameters $v^*$ from $d$ queries $(v^*z_1, \ldots, v^*z_d)$ by choosing, for

$i \in [\![1, d]\!]$, $z_i = (\delta_{ij})_{j \in [\![1, n]\!]}$, effectively probing each coordinate individually. This leads to the maximum possible information leakage allowed by the post-processing upper bound. Strikingly, we now show that for some datasets, the adversary can in fact induce this maximal privacy leakage *with just one query*.

By definition, $Wz = Vz + \mu z$ and therefore $Wz$ is the result of shifting the distribution $Vz$ by $\mu z$. The norm of the shift between $Wz$ and $Vz$ is thus $\|\mu z\|$, while the norm of the shift between $V$ and $W$ is $\|\mu\|$. We can compare, for a given shift $\mu$ and a given seed $z$, the trade-off functions of $(V, W)$ and $(Vz, Wz)$. We have $T(Vz, Wz) = G_{\|\mu z\|/\|z\|\sigma_\theta}$. For a given shift $\mu$ between $V$ and $W$, which is known by the adversary in the threat model of DP, the adversary can maximize $\|\mu z\|/\|z\|$ by taking $z$ to be the right singular vector corresponding to the largest singular value $\sigma_{\max}(\mu)$ of $\mu$. We directly obtain that

$$T(Vz, Wz) = G_{|\sigma_{\max}(\mu)|/\sigma_\theta}.$$

Hence, for a hypothesis test between a particular instantiation of shift $\mu \in \mathbb{R}^{n \times d}$, the sensitivity of the exact trade-off function between $Vz$ and $Wz$ is improved from $\|\mu\|/\sigma_\theta$ to $|\sigma_{\max}(\mu)|/\sigma_\theta$. However, these considerations do not imply better privacy guarantees than releasing the model parameters, as shown by the following result.

**Proposition 3.1.** *For any fixed $z \in \mathbb{R}^d$, there exist adjacent datasets $\mathcal{D}$ and $\mathcal{D}'$ such that:*

$$T(Vz, Wz) = T(V, W).$$

In other words, for any possible query $z \in \mathbb{R}^d$, there exists two (pathological) adjacent datasets $\mathcal{D}, \mathcal{D}'$ such that performing the query $\mathcal{M}(\mathcal{D})z$ implies the same privacy leakage as directly releasing $\mathcal{M}(\mathcal{D})$.

Note however that our results show that, for a specific shift $\mu \in \mathbb{R}^{n \times d}$, the sensitivity of the exact trade-off function between $Vz$ and $Wz$ is $|\sigma_{\max}(\mu)|/\sigma_\theta$, which can be strictly smaller than the norm-based bound $\|\mu\|/\sigma_\theta$. This indicates that, from an empirical standpoint, the actual privacy leakage may be lower than the worst-case upper bound implied by the post-processing theorem for realistic datasets.

We provide the detailed proof and a discussion about the choice of $z$ in Appendix A.1.

### 3.2. Noisy Gradient Descent

We now extend the previous results to the case where the private generative model is trained with Noisy Gradient Descent (NGD) (Song et al., 2013; Bassily et al., 2014; Abadi et al., 2016; Feldman et al., 2018; Altschuler and Talwar, 2022).

Specifically, we present negative results in the context of Label Differential Privacy (Label DP), where adjacent datasets

only differ in the labels (Ghazi et al., 2021). Formally, adjacent datasets under label DP can be written $\mathcal{D} = (X, Y)$, $\mathcal{D}' = (X, Y')$, where $Y \in \mathbb{R}^{n \times m}$ and $Y' \in \mathbb{R}^{n \times m}$ differ in exactly one column (we say $Y, Y'$ are adjacent). Since any two datasets that are adjacent under Label DP remain adjacent under standard DP (where both features and labels can differ), the maximum privacy leakage under standard DP is at least as large. Our negative results for Label DP thus extend to standard DP.

Let $V_0 = W_0 \in \mathbb{R}^{n \times d}$ a standard Gaussian initialization, $F(w, X, Y) = \frac{1}{m} \sum_{k=1}^{m} \|wx_k - y_k\|^2 + \lambda \|w\|^2$ the objective function, $\eta > 0$, $\sigma > 0$ and $\{N_t\}_t$ a sequence of i.i.d standard Gaussian matrices of $\mathbb{R}^{n \times d}$. On two adjacent datasets $\mathcal{D} = (X, Y), \mathcal{D}' = (X, Y')$, NGD corresponds to the following updates:

$$V_{t+1} = V_t - \eta \nabla_w F(V_t, X, Y) + \sqrt{2\eta}\sigma N_{t+1},$$
$$W_{t+1} = W_t - \eta \nabla_w F(W_t, X, Y') + \sqrt{2\eta}\sigma N_{t+1}.$$

Note that $V_t$ can be decomposed into independent Gaussian rows, and likewise for $W_t$.

In the following, we focus on the case where the model is trained until convergence (i.e., $t \to \infty$) before querying it to release synthetic data (a finite-time analysis can also be done, see Appendix A.2). Our objective is thus to compare the trade-offs functions $T(V_\infty, W_\infty)$—for releasing the model directly—and $T(V_\infty z, W_\infty z)$—for releasing the output of the model on a query $z \in \mathbb{R}^d$.

As a discretized Langevin dynamical system with a strongly convex, smooth objective, it is known that $V_t$ converges in distribution to a limit distribution (Durmus et al., 2019), which is the Gibbs stationary distribution when $\eta \to 0$. We can leverage this consideration to characterize $T(V_\infty, W_\infty)$ and $T(V_\infty z, W_\infty z)$.

**Proposition 3.2.** *Let $\Sigma = \frac{1}{n}X^T X + \lambda I$, $M = I - 2\eta\Sigma$ and denote by $A$ the square root of $\Sigma^{-1}M$ and $B$ the square root of $\Sigma^{-1}M^{-1}$. Assume that $Y$ and $Y'$ are adjacent and that $\eta(\lambda + M_x^2/n) < 1$. Then:*

$$T(V_\infty, W_\infty) = G_{\frac{\|AX^T(Y-Y')\|}{n\sigma}},$$
$$T(V_\infty z, W_\infty z) = G_{\frac{\|z^T \Sigma^{-1} X^T (Y-Y')\|}{n\sigma \|Bz\|}}.$$

As in the output perturbation setting, the adversary aims to maximize the privacy leakage from a single data point. Thus, her objective is, for a fixed pair of adjacent datasets $\mathcal{D}, \mathcal{D}'$ to find $\sup_{z \in \mathbb{R}^d} T(V_\infty z, W_\infty z)$. The following proposition quantifies the associated privacy leakage.

**Proposition 3.3.** *For any pair of adjacent datasets (in the label DP sense), the adversary can choose $z \in \mathbb{R}^d$ such that:*

$$T(V_\infty z, W_\infty z) = T(V_\infty, W_\infty).$$

This statement gives a negative result similar to the one obtained for output perturbation (Proposition 3.1): releasing $V_\infty z$ does not offer any privacy amplification compared to releasing the model $V_\infty$ in the worst case.

We provide the detailed proofs of the above results and a discussion about the choice of $z$ in Appendix A.3.

# 4. Privacy Amplification for Releasing Synthetic Data from Random Inputs

Motivated by the negative results of the previous section, we now consider the case where synthetic data points are generated by feeding random inputs into a linear regression model privatized via output perturbation. This relaxed threat model reflects the common scenario where a trusted party trains the generative model, generates $l$ synthetic data points, and releases only these points—without revealing the generative model or the random inputs used in the generation process.

Remarkably, we demonstrate that releasing synthetic data points provides stronger privacy guarantees than directly releasing the model when $l \ll d$, highlighting the key role of randomization in the privacy of synthetic data generation.

## 4.1. Setting

As before, we consider the linear regression setting introduced in Section 2.2. We privatize the model via output perturbation, as in Section 3.1, which we recall is given by $\mathcal{M}(\mathcal{D}) = \arg\min_{w \in \mathbb{R}^{n \times d}} F_\lambda(w; \mathcal{D}) + \sigma_\theta N$, where $N_{ij} \overset{iid}{\sim} \mathcal{N}(0, 1)$. The difference is that the seeds $Z \in \mathbb{R}^{d \times l}$ used to generate $l$ synthetic data points are now Gaussian. More precisely, we consider the following mechanism.

**Definition 4.1** (Synthetic data generation from random inputs). Let $\mathcal{D}$ be a dataset and $\mathcal{M}$ denote the output perturbation mechanism. Our objective is to analyze the privacy guarantees of the mechanism $\mathcal{M}_Z(v) = \mathcal{M}(v)Z$, where $Z \in \mathbb{R}^{d \times l}$ with $Z_{ij} \overset{iid}{\sim} \mathcal{N}(0, \sigma_z^2)$.

For two adjacent datasets $\mathcal{D}$ and $\mathcal{D}'$, we define $v^* = \arg\min_{w \in \mathbb{R}^{n \times d}} F_\lambda(w; \mathcal{D})$ and $w^* = \arg\min_{w \in \mathbb{R}^{n \times d}} F_\lambda(w; \mathcal{D}')$.

The post-processing theorem (Dong et al., 2022) ensures that we have:

$$T(\mathcal{M}_Z(v^*), \mathcal{M}_Z(w^*)) \geq T(\mathcal{M}(v^*), \mathcal{M}(w^*)).$$

However, the post-processing theorem may not be tight for this mechanism, i.e., equality may not hold in the inequality above. In order to assess the potential privacy amplification phenomenon associated with this mechanism, we aim to compute, or at least estimate, $T(\mathcal{M}_Z(v^*), \mathcal{M}_Z(w^*))$.

Throughout the rest of the section, we let $v, w \in \mathbb{R}^{n \times d}$ such that $\|v - w\| \leq \Delta$. We note $V = \mathcal{M}(v) = v + \sigma_\theta N$ and $W = \mathcal{M}(w) = w + \sigma_\theta N$.

$VZ$ and $WZ$ are two distributions of a family that can be parametrized by the mean of the left Gaussian matrix in the product ($v$ and $w$). We denote as $P_v$ the distribution of $VZ$ and $P_w$ the distribution of $WZ$. The trade-off function between $V$ and $W$ is equal to:

$$T(V, W)(\alpha) = \Phi(\Phi^{-1}(1 - \alpha) - \|w - v\| / \sigma_\theta),$$

where $\Phi$ is the c.d.f of a standard Gaussian variable.

We can derive closed-form expressions for $P_v$, as shown in the following lemma.

**Lemma 4.1.** (Distribution of $VZ$ and $WZ$). *The distribution $P_v$ has the following characteristic function:*

$$\phi_{P_v}(t) = \frac{\exp\left(\frac{-\sigma_z^2}{2} \operatorname{tr}(t^T vv^T t(I_l + \sigma_z^2 \sigma_\theta^2 t^T t)^{-1})\right)}{\det\left(I_l + \sigma_z^2 \sigma_\theta^2 t^T t\right)^{d/2}}.$$

The proof can be found in Appendix B.1. For $v = 0$ and $l = 1$, the distribution $P_v$ corresponds to a generalized Laplace distribution, which has the following density (Mattei, 2017):

$$P_0(s) = \frac{2}{\sqrt{\pi^n (2\sigma_z \sigma_\theta)^{n+d}}} \frac{\|s\|^{\frac{d-n}{2}}}{\Gamma(d/2)} K_{\frac{d-1}{2}}\left(\frac{\|s\|}{\sigma_z \sigma_\theta}\right),$$

where $\pi$ is a constant, $\Gamma$ is the Gamma distribution and $K_{\frac{d-1}{2}}$ is the modified Bessel function of the second kind of order $\frac{d-1}{2}$. However, to our knowledge, when $v \neq 0$, the distribution $P_v$ does not correspond to any standard or well-studied distribution, and its density lacks a simple closed-form expression (Li and Woodruff, 2021).

## 4.2. Releasing a Single Point

The above observations suggest that the exact computation of $T(VZ, WZ)$ or $D_\alpha(VZ, WZ)$ is likely intractable, if not impossible in general. In this section, we focus on the simplest case, where $n = l = 1$, and $d \gg 1$. In other words, the input dimension is large, and we release a single one-dimensional synthetic data point. In this special case, we can derive simple, non-asymptotic privacy bounds by leveraging a univariate variation of the Central Limit Theorem (CLT). Indeed, $VZ \in \mathbb{R}$, so we avoid complications related to dimensionality that arise in the multivariate CLT. As a result, we obtain tighter bounds with better scaling in $d$ than would be possible in the higher-dimensional case $(n, l > 1)$. In contrast, multivariate non-asymptotic CLTs break down when the dimension of the random vector becomes large—we discuss the applicability of our results in higher dimensions in Section 4.3.

Specifically, we observe that $VZ$ can be decomposed into a sum of independent terms: $VZ = \sum_{k=1}^{d} V_k Z_k$, with $Z \sim \mathcal{N}(0, \sigma_z^2 I_d)$. From there, we apply a central limit result to approximate $VZ$ and $WZ$ by Gaussian variables for which the trade-off function is known. The following lemma is essential to our reasoning.

**Lemma 4.2.** *(Approximate trade-off function).* *Let $P, Q, \tilde{P}, \tilde{Q}$ four distributions of $\mathbb{R}^d$. Let $\gamma = \max(TV(\tilde{P}, P), TV(\tilde{Q}, Q))$ where $TV$ denotes the total variation between distributions. Let $\alpha \in (\gamma, 1 - \gamma)$. Then,*

$$T(\tilde{P}, \tilde{Q})(\alpha + \gamma) - \gamma \leq T(P, Q)(\alpha) \leq T(\tilde{P}, \tilde{Q})(\alpha - \gamma) + \gamma.$$

We refer to Appendix B.2 for the proof. This lemma ensures that the non-asymptotic bounds of the CLT translate into bounds for the trade-off function. Moreover, we use the following theorem derived from (Bally and Caramellino, 2016) to establish convergence to the Gaussian distribution.

**Theorem 4.2** (Multivariate CLT asymptotic development in total variation distance (informal, adapted from Theorem 2.6. of Bally and Caramellino, 2016))**.** *Let $F = \{F_k\}_k$ be a sequence of i.i.d random variables in $\mathbb{R}^N$ absolutely continuous with respect to the Lebesgue measure, with null mean and invertible covariance matrix $C(F)$. Let $G \sim \mathcal{N}(0, I_N)$. Let $A(F) = C(F)^{-1/2}$ and $S_d = \frac{1}{\sqrt{d}} \sum_{k=1}^{d} A(F) F_k$. Let $r \geq 2$. If $\mathbb{E}[|F_1|^{r+1}] < +\infty$ and all moments up to order $r$ of $A(F)F_1$ agree with the moments of a standard Gaussian r.v. in $\mathbb{R}^N$, then:*

$$TV(S_d, G) \leq C(1 + \mathbb{E}[|F_1|^{r+1}])^{\max\{r/3, 1\}} \times \frac{1}{d^{(r-1)/2}},$$

*where $C$ depends on $r$, $N$ and $C(F)$.*

In the univariate case, this theorem takes a particularly simple form given below.

**Lemma 4.3.** *Let $G \sim \mathcal{N}(0, 1)$. Then, there exists $A_{\|v\|} > 0$ such that:*

$$TV\left(\sqrt{d}(\sigma_\theta N + v)Z, \sigma_z \sqrt{d\sigma_\theta^2 + \|v\|^2} G\right) \leq \frac{A_{\|v\|}}{d}.$$

We have established that $VZ$ and $WZ$ can be approximated by Gaussian variables, both with zero mean but different variance. In the univariate case and for large $d$, our synthetic data release thus transforms a mean shift between $V$ and $W$ as described by the relationship $W = V + w - v$ into a variance shift. The variance shift is captured by the approximation:

$$WZ \approx \sqrt{\frac{d\sigma_\theta^2 + \|w\|^2}{d\sigma_\theta^2 + \|v\|^2}} VZ \text{ for } d \gg 1.$$

For conciseness, for $x, y > 0$, we denote by $\Lambda(\sigma_\theta, d, x, y)$ the pair $\left(\sqrt{d\sigma_\theta^2 + x^2}, \sqrt{d\sigma_\theta^2 + y^2}\right)$. Now, we need to compute the trade-off function of these Gaussian variables.

**Proposition 4.1.** (Trade-off function between Gaussians with different variance). *Let $\sigma_1, \sigma_2 > 0$. Then,*

$$T(\mathcal{N}(0, \sigma_1^2), \mathcal{N}(0, \sigma_2^2)) = \begin{cases} T_1(\alpha) \text{ if } \sigma_1 \leq \sigma_2, \\ T_2(\alpha) \text{ else,} \end{cases}$$

*where* $\quad T_1(\alpha) = 2\Phi\left(\frac{\sigma_1}{\sigma_2}\Phi^{-1}(1 - \alpha/2)\right) - 1,$

$$T_2(\alpha) = 2 - 2\Phi\left(\frac{\sigma_1}{\sigma_2}\Phi^{-1}((\alpha + 1)/2)\right).$$

*We denote this trade-off function by $\tilde{G}_{(\sigma_1, \sigma_2)}$.*

We refer to Appendix B.4 for the proof. With these results, we can approximate the trade-off function of interest (corresponding to linear regression in the univariate output setting) by that of Gaussians, as stated in the following theorem.

**Theorem 4.3.** *Let $d > 0$. Then, there exists a universal constant $C > 0$ such that for all $\alpha \in (C/d, 1 - C/d)$:*

$$\tilde{G}_{\Lambda(\sigma_\theta, d, \|v\|, \|w\|)}\left(\alpha + \frac{C}{d}\right) - \frac{C}{d} \leq T(VZ, WZ)(\alpha),$$

$$\tilde{G}_{\Lambda(\sigma_\theta, d, \|v\|, \|w\|)}\left(\alpha - \frac{C}{d}\right) + \frac{C}{d} \geq T(VZ, WZ)(\alpha).$$

This theorem states that the trade-off function between $VZ$ and $WZ$ converges in $\mathcal{O}(1/d)$ to a trade-off function between two Gaussian variables with different variances. Combining our bounds with the post-processing theorem, we have shown that:

$$T(VZ, WZ) \geq \max \begin{cases} T(V, W), \\ \tilde{G}_{\Lambda(\sigma_\theta, d, \|v\|, \|w\|)}\left(\cdot + \frac{C}{d}\right) - \frac{C}{d}. \end{cases}$$

Below, we will denote this lower bound by $h$. Figure 1 represents a numerical computation of the upper bound.

**Interpretation with Rényi divergences.** Translating the convergence of the trade-off function into a formal privacy amplification bound is nontrivial. We can first examine the Rényi divergence between the limiting Gaussian distributions. We denote $\nu_v^d = \mathcal{N}(0, \sigma_z^2(d\sigma_\theta^2 + v^2))$. The Rényi divergence is maximized for $v = v_*, w_* = v_* + \Delta$ for some value $v_*$ which depends on $d, \Delta$ and $\sigma_\theta$ (more details in Appendix B.7). Then, it can be shown that:

$$D_\alpha(\nu_{v_*}^d, \nu_{w_*}^d) = \frac{\alpha \Delta^2}{4d\sigma_\theta^2} + o(d^{-1}).$$

This result yields two key insights. First, as $d \to \infty$, $\nu_{v_*}^d$ is indistinguishable from $\nu_{w_*}^d$, supporting the idea that $VZ$ becomes indistinguishable from $WZ$. Second, we have:

$$D_\alpha(\nu_{v_*}^d, \nu_{w_*}^d) \approx \frac{1}{2d} D_\alpha(V, W).$$

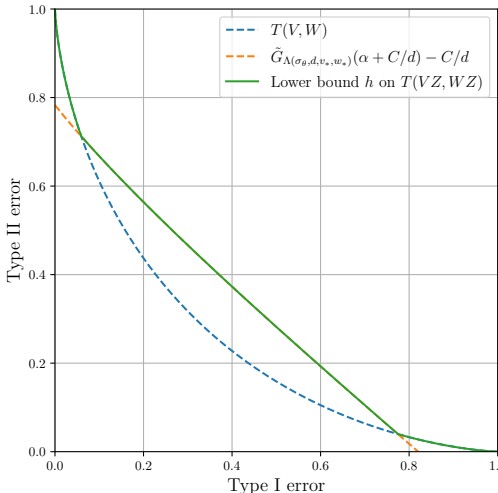

*Figure 1.* Comparison of the lower bound $h$ on $T(VZ, WZ)$ to $T(V, W)$ and $\tilde{G}_{\Lambda(\sigma_\theta, d, \Delta)}\left(\alpha + \frac{C}{d}\right) - \frac{C}{d}$ for $C = 1, \Delta = 1, \sigma_\theta = 1, d = 12$.

This comparison with the post-processing upper bound $D_\alpha(V, W)$ suggests a privacy amplification of order $\mathcal{O}(1/d)$ in the asymptotic regime. However, this observation alone does not suffice to conclude that $D_\alpha(VZ, WZ) = \mathcal{O}(1/d)$. The difficulty arises from the fact that the $\mathcal{O}(1/d)$ convergence rate of the lower bound $h$ on the trade-off function does not directly imply a corresponding convergence rate for the Rényi divergence between the original (non-Gaussian) distributions.

We thus numerically approximate $D_\alpha(VZ, WZ)$ using $h$ to gain more intuition. This is done via the following conversion result from Dong et al. (2022).

**Proposition 4.2** (Conversion from $f$-DP to RDP (Dong et al., 2022)). *If a mechanism is $f$-DP, then it is $(\alpha, l_\alpha(f))$-RDP for all $\alpha > 1$ with:*

$$l_\alpha(f) = \begin{cases} \frac{1}{\alpha-1} \log \int |f'(t)|^{1-\alpha} dt \text{ if } z_f = 1, \\ +\infty \text{ else,} \end{cases}$$

*with $z_f = \inf\{t \in (0, 1); f(t) = 0\} = 1$.*

While it is difficult to theoretically compute $h$, for $d$ large enough, there exist $0 < c_1 < c_2 < 1$ such that:

$$h(\alpha) = \begin{cases} T(V, W)(\alpha) \text{ if } \alpha \in (0, c_1) \cup (c_2, 1), \\ \tilde{g}(\alpha) \text{ if } \alpha \in (c_1, c_2), \end{cases}$$

where $\tilde{g}(\alpha) = \tilde{G}_{\Lambda(\sigma_\theta, d, \Delta)}\left(\alpha + \frac{C}{d}\right) - \frac{C}{d}$. This matches the numerical representation in Figure 1. Working with $h$ rather than $\tilde{g}$ alone is essential, because $z_{\tilde{g}} < 1$ implies $l_\alpha(\tilde{g}) = +\infty$, yielding no meaningful RDP guarantee. In contrast, $l_\alpha(h)$ remains finite, allowing us to derive valid bounds.

For multiple values of $d$ and $\Delta$, we first determine $c_1$ and $c_2$. Then, we estimate $l_\alpha(h)$ using a Monte Carlo approximation:

$$l_\alpha(h) \approx \frac{1}{L(\alpha-1)} \log \sum_{k=1}^{L} |h'(X_k)|^{1-\alpha},$$

where $(X_k)_{k \geq 1} \overset{iid}{\sim} \text{Unif}([0, 1])$ and $L = 10^6$ is the number of samples. We run the procedure $M = 50$ times, and the estimates are averaged. Figure 2 reports the logarithm of the estimated divergence as a function of $\log(d)$ for multiple values of $\Delta$. Standard deviations are not shown as they are negligible compared to the estimated values. The results suggest a convergence rate $l_\alpha(h) \approx \mathcal{O}(1/d)$ in the high privacy regime $\Delta < 1$. The initial plateau observed for small $d$ arises from limitations in our analysis. Specifically, it reflects conservative constants in Lemma 4.3 and the worst case-approximation of trade-off functions of Theorem 4.3, rather than an intrinsic property of the divergence itself. More details about the experiments can be found in Appendix B.7.

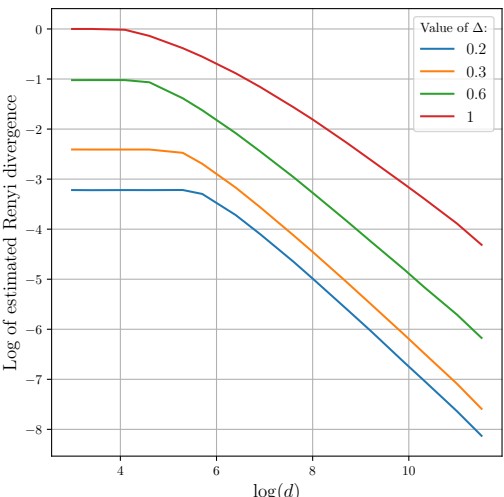

*Figure 2.* Privacy amplification in Rényi DP when releasing a single point: estimation of $D_\alpha(VZ, WZ)$ as a function of $\log(d)$ for different values of $\Delta$.

### 4.3. Releasing Multiple Points

We now consider the more general case where $l \geq 1$ synthetic data points in dimension $n \geq 1$ are released. In this case, the adversary can leverage correlations between the different outputs to better reconstruct $V$ or $W$ and thus yield higher privacy leakage. In fact, the elements of $VZ$ are not independent. This means that we cannot directly leverage the results of the previous section and apply composition theorems to obtain an approximation of the trade-off function $T(VZ, WZ)$. However, $VZ$ can be vectorized and

decomposed into a sum of independent vectorized matrices: $\text{vec}(VZ) = \sum_{k=1}^{d} \text{vec}(V_{.,k}Z_k)$, where $V_{.,k}$ is the $k$-th column of $V$. Then, we can apply Theorem 4.2 for random vectors and expect the distribution of $\text{vec}(VZ) \in \mathbb{R}^{nl}$ to converge to a Gaussian. However, the constants in Theorem 4.2 depend on the dimension of $\text{vec}(VZ)$, here $N = nl$. To our knowledge, no multivariate central limit theorem currently provides total variation distance bounds with explicit constants yet. Raič (2019) studied Berry-Esseen bounds for the central limit theorem and obtained universal constants equal to $42N^{1/4} + 16$, which become uninformative when $N$ is larger than some power of $d$.

A recent work solved this issue in the special case of the product of i.i.d Gaussian matrices.

**Theorem 4.4** (Convergence of product of Gaussian matrices (adapted from Theorem 1 of (Li and Woodruff, 2021)). *Suppose that $d \geq \max\{n, l\}$ and $N \in \mathbb{R}^{n \times d}, Z \in \mathbb{R}^{d \times l}, G \in \mathbb{R}^{n \times l}$ are standard Gaussian matrices. Then, there exists $C > 0$ such that:*

$$TV(NZ, \sqrt{d}G) \leq C\sqrt{\frac{nl}{d}}.$$

This theorem must be adapted to our case, where $N$ is shifted by some matrix $v$. The following theorem establishes convergence even the presence of a shift $v$ that induces dependence between $NZ$ and $vZ$, complicating the analysis. We provide a non asymptotic upper bound on the TV distance between $(\sigma_\theta N + v)Z$ and a Gaussian matrix. Strikingly, our bound does not depend on the norm of the shift $v$.

**Theorem 4.5.** (Convergence of product of Gaussian matrices, shifted version). *Let $N \in \mathbb{R}^{n \times d}, Z, Z' \in \mathbb{R}^{d \times l}, G \in \mathbb{R}^{n \times l}$ be independent standard Gaussian matrices. Let $s = \text{rank}(v)$. Assume that $d \geq \max\{n, l\}$. Then, there exists $C' > 0$ such that:*

$$TV\left((\sigma_\theta N + v)Z, \sigma_\theta\sqrt{d-s}G + vZ'\right) \leq C'\sqrt{\frac{nls}{d-s}}.$$

*Sketch of proof (details in Appendix B.6):* For simplicity, we set $\sigma_\theta = 1$. The idea of the proof is to use the SVD of $v = F\Sigma S^T$ and use the invertibility of $F$ and $S$ and the invariance of Gaussian matrices by orthogonal transformation to write $TV((N + v)Z, \sqrt{d-s}G + vZ') = TV((N + \Sigma)Z, \sqrt{d-s}G + \Sigma Z')$ and observe that for $d > \max\{n, l\}$, $\Sigma = (\lambda_1, \ldots, \lambda_s, \underbrace{0, \ldots, 0}_{n-s \text{ times}})$.

Then, we decompose $(N + v)Z$ into a part that depends on the shift $v$ and vanishes as $d \to +\infty$ and another part that we can approximate with a Gaussian matrix: $H = \sum_{k=1}^{s}(N_{.,k} + \Sigma_{.,k})Z_k$ and $N_{-s}Z_{-s} = \sum_{k=s+1}^{d} N_{.,k}Z_k$, and $NZ = N_{-s}Z_{-s} + H$. Note that $N_{-s}Z_{-s}$ and $H$ are independent.

By Theorem 4.4, we can approximate $N_{-s}Z_{-s}$ as a Gaussian matrix $\sqrt{d-s}G$. Using the triangle inequality for the TV distance:

$$TV\left(N_{-s}Z_{-s} + H, \sqrt{d-s}G + vZ'\right)$$
$$\leq TV\left(N_{-s}Z_{-s} + H, \sqrt{d-s}G + H\right) \quad (1)$$
$$+ TV\left(\sqrt{d-s}G + H, \sqrt{d-s}G + \Sigma Z'\right). \quad (2)$$

Using independence in the decomposition and the post-processing theorem, we have:

$$(1) \leq TV\left((N_{-s}Z_{-s}, H), (\sqrt{d-s}G, H)\right) \leq C\sqrt{\frac{nl}{d-s}}.$$

Also, (2) can be seen as the total variation distance between the convolution of two distributions by a third one. We use Pinsker inequality to relate this TV distance to a KL divergence. Then, we define a random variable $W$ and write: $G + H = G + W + H - W$. Using the post-processing inequality, the chain rule and convexity for KL divergences, we obtain:

$$D_{KL}(\sqrt{d-s}G + \Sigma Z', \sqrt{d-s}G + H)$$
$$\leq D_{KL}(\Sigma Z' + W, H)$$
$$+ \mathbb{E}_{w \sim W}[D_{KL}(\sqrt{d-s}G - w, \sqrt{d-s}G)].$$

Setting $W = \sum_{k=1}^{s} N_{.,k}Z'_k$, we get $D_{KL}(\Sigma Z' + W, H) = 0$ and:

$$D_{KL}(\sqrt{d-s}G + \Sigma Z', \sqrt{d-s}G + H) \leq \frac{nls}{2(d-s)}.$$

This concludes the proof. $\qquad\square$

This theorem allows us to recover privacy amplification results in the multiple outputs scenario. By abuse of notation, for $G \in \mathbb{R}^{n \times l}$ a Gaussian matrix independent of $Z$, we denote $\tilde{G}_{(\sigma_\theta, d-n, v, w)} = T(\sigma_\theta\sqrt{d-n}G + vZ, \sigma_\theta\sqrt{d-n}G + wZ)$.

**Theorem 4.6.** *Let $d > 0$. Then, there exists a universal constant $C' > 0, C_{n,l,d} = C'n\sqrt{\frac{l}{d-n}}$ such that for all $\alpha \in (C_{n,l,d}, 1 - C_{n,l,d})$:*

$$\tilde{G}_{(\sigma_\theta, d-n, v, w)}(\alpha + C_{n,l,d}) - C_{n,l,d} \leq T(VZ, WZ)(\alpha)$$
$$\tilde{G}_{(\sigma_\theta, d-n, v, w)}(\alpha - C_{n,l,d}) + C_{n,l,d} \geq T(VZ, WZ)(\alpha).$$

Combining our bounds with the post-processing theorem and denoting $\tilde{g}(\alpha) = \tilde{G}_{(\sigma_\theta, d-n, v, w)}(\alpha + C_{n,l,d}) - C_{n,l,d}$, we have shown that:

$$T(VZ, WZ) \geq \max\{T(V, W), \tilde{g}\}.$$

We note $h$ this upper bound, with a slight abuse of notation.

**Interpretation with Rényi divergences.** Leveraging Rényi divergences allows us to interpret our result both as a form of privacy amplification via synthetic data release and as a composition theorem for the "release one point" mechanism of Section 4.2, which is valid when only a small number of synthetic points are released, i.e., $\max\{n, l\} \leq d$.

Similar to the case $l = n = 1$, we can derive Rényi divergence privacy bounds for the limiting distributions in order to get some intuition about the relationship between the parameters and the privacy loss. Denoting $G_v = \sigma_\theta \sqrt{d - n}G + vZ, G_w = \sigma_\theta \sqrt{d - n}G + wZ, D_\alpha(G_v, G_w)$ can be upper bounded as follows (see Appendix B.7):

$$D_\alpha(G_v, G_w) \leq \frac{\alpha n l \Delta^2}{4(d - n)\sigma_\theta^2} + o(d^{-1}).$$

Analogous to Section 4.2, $G_v$ and $G_w$ converge in distribution when $d \to +\infty$. However, this time, there is a dependence in $nl$ which behaves as a composition result of the "release one point" mechanism. Comparing to the post-processing upper bound $D_\alpha(V, W)$, we now have:

$$D_\alpha(G_v, G_w) \lesssim \frac{nl}{2(d - n)} D_\alpha(V, W).$$

Following the same procedure described in Section 4.2, we numerically estimate the Rényi divergence between $VZ$ and $WZ$ for various values of $d$. Figure 3 suggests an approximate convergence rate of $\mathcal{O}(d^{-1/2})$ when $l$ and $n$ are fixed. More details about the experiments can be found in Appendix B.7.

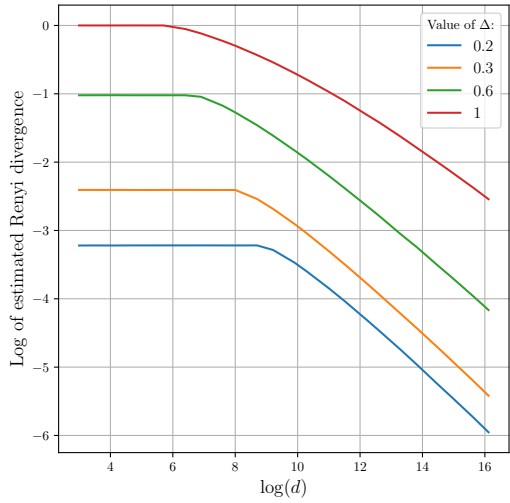

*Figure 3.* Privacy amplification in Rényi DP when releasing multiple points ($l = 10, n = 1$): estimation of $D_\alpha(VZ, WZ)$ as a function of $\log(d)$ for different values of $\Delta$.

**Discussion.** When $d$ is sufficiently large, the privacy loss incurred by releasing $nl$ synthetic data points is lower than

that of directly releasing the model parameters. The resulting guarantees are comparable in order to those obtained by training the model non-privately and releasing $nl$ noisy predictions—each individually satisfying differential privacy. However, as discussed in Section 1, directly adding noise to model outputs is undesirable, as it requires difficult and often loose sensitivity analyses and often degrade utility. In contrast, although we privatize the model itself, we obtain bounds that resemble the composition of the "release one point" mechanism. It is important to note, however, that this convergence analysis holds only when $d \geq \max\{n, l\}$.

We believe that this behavior may hold in more general settings, but our proof techniques heavily rely on the convergence theorem of products of Gaussians. Unfortunately, there is no hope to apply this technique in the "small $d$" regime, as proven by Li and Woodruff (2021).

**Theorem 4.7** (Non convergence of Gaussian product for small $d$, adapted from Theorem 1 in Li and Woodruff, 2021)**.**
*Let $G_1 \in \mathbb{R}^{n \times d}, G_2 \in \mathbb{R}^{d \times l}, G \in \mathbb{R}^{n \times l}$ be Gaussian matrices. Suppose that*

$$d \leq C'' \max\{n, l\}^{1/2} \min\{n, l\}^{3/2},$$

*with $C''$ an universal constant. Then:*

$$TV(G_1 G_2, \sqrt{d}G) \geq 2/3.$$

## 5. Conclusion & Perspectives

We have shown that there exists a privacy amplification phenomenon for synthetic data in the context of linear regression. However, there is no amplification when the adversary has control over the seed of the synthesizer.

This negative result could inform the development of tighter privacy auditing strategies for synthetic data release (Annamalai et al., 2024). By quantifying the degradation in privacy guarantees, our findings offer insights that can help design more robust auditing methods in adversarial settings.

Several important directions remain for future work, including deriving general privacy amplification bounds that hold when $n, l$ are large or $d$ is small. A complete investigation of privacy bounds for more advanced privacy preserving algorithms such as Projected Noisy Gradient Descent or DP-SGD, for fixed and random output settings, is also essential. Finally, extending our analysis to more complex models such as neural networks is a crucial step toward making these theoretical results applicable to real-world scenarios.

## Impact Statement

This paper presents work whose goal is to advance privacy in machine learning, offering tools to make it more secure. Private synthetic data generation is recognized as a promising approach for enabling flexible yet privacy-preserving

analyses. Studying privacy amplification phenomena in this context is valuable because it allows for tighter quantification of privacy guarantees, thereby strengthening protection. Moreover, it assists data curators in designing more effective privacy-preserving algorithms.

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

This appendix presents detailed proofs and additional discussion.

## A. Proofs of Section 3

### A.1. Proof of Proposition 3.1

**Proposition 3.1.** *For any fixed $z \in \mathbb{R}^d$, there exist adjacent datasets $\mathcal{D}$ and $\mathcal{D}'$ such that:*

$$T(Vz, Wz) = T(V, W).$$

*Proof.* We simply write $Wz = Vz + \mu z$. Then, $T(Vz, Wz) = G_{\|\mu z\|/\|z\|\sigma_\theta}$. Then, $\|\mu z\|/\|z\|$ is maximized by taking $z$ the right singular vector corresponding to the largest singular value of $\mu$.

Now, we prove that for all $z \in \mathbb{R}^d$, there exists $\mathcal{D}, \mathcal{D}'$ such that $T(Vz, Wz) = T(V, W)$. In order to do this, we prove that there exists two adjacent datasets $\mathcal{D} = (X, Y)$, $\mathcal{D}' = (X, Y')$ such that the shift $\mu$ between $V$ and $W$, the weights matrices of linear regression on $\mathcal{D}$ and $\mathcal{D}'$ at convergence, has rank 1.

Let $a \in \mathbb{R}^m, u = z/\|z\| \in \mathbb{R}^d, v \in \mathbb{R}^n$. Let $X = au^T \in \mathbb{R}^{m \times d}, Y = av^T \in \mathbb{R}^{m \times n}$, be two rank one matrices. Therefore, $X^T Y = (a^T a)uv^T$ is rank one. We know that regularized linear regression converges to $Y^T X(X^T X + \lambda I)^{-1}$. By Sherman-Morrison formula,

$$(X^T X + \lambda I)^{-1} = (\|a\|^2 uu^T + \lambda I)^{-1} = \frac{1}{\lambda}I - \frac{1}{\lambda}\frac{uu^T}{\lambda + \|u\|^2}.$$

Then,

$$Y^T X(X^T X + \lambda I)^{-1} = \frac{\|a\|^2}{\lambda}vu^T - \frac{\|u\|^2 \|a^2\|}{\lambda}\frac{vu^T}{\lambda + \|u\|^2}.$$

Finally, choosing $Y'$ proportional to $Y$, the shift has rank 1 and $\|\mu\| = |\sigma_{\max}(\mu)| = \|\mu z\|/\|z\|$. □

### A.2. Discussion on finite training time and convergence of NGD

We prove that full batch NGD converges to a normal distribution if $\eta(\lambda + M_x^2/n) < 1$.

**Proposition A.1.** *Let $t \in \mathbb{N}^*$. Then, $V_T$ is composed of independent Gaussian columns $(V_t)_i \overset{iid}{\sim} \mathcal{N}(\mu_t^i, \Sigma_t)$ with:*

$$\mu_t^i = \sum_{k=0}^{t-1} B_i M^k = B_i(I - M)^{-1}(I - M^t)$$

$$\Sigma_t^i = M^{2t} + \eta\sigma^2 \sum_{k=0}^{t-1} M^{2k} = M^{2m} + \eta\sigma^2(I - M^2)^{-1}(I - M^{2t}).$$

*Proof.* The gradient is equal to

$$\nabla_w F_\lambda(V_k, X, Y) = \frac{1}{n}\sum_{i=1}^m \nabla_w f(V_k, x_i, y_i) + 2\lambda V_k = \frac{2}{n}(V_k X^T - Y^T)X + 2\lambda V_k.$$

Then, noting $B = \frac{2\eta}{n}Y^T X, \Sigma = \frac{1}{n}X^T X + \lambda I$ and $M = I - 2\eta\Sigma$, we get:

$$V_{k+1} = V_k - \eta\nabla_w F_\lambda(V_k, X, Y) + \sqrt{\eta}N_{n+1} = V_k(I - 2\eta(X^T X/n + \lambda I)) - \frac{2}{n}Y^T X = V_k M + B + \sqrt{2\eta}\sigma N_{k+1}.$$

Then, we can write:

$$V_t = V_0 M^t + \sum_{k=0}^{t-1}(B + \sqrt{2\eta}\sigma N_{k+1})M^{t-1-k},$$

which is composed of independent lines with mean and covariance:

$$\mu_t^i = \sum_{k=0}^{t-1} B_i M^k = B_i (I - M)^{-1}(I - M^t),$$

$$\Sigma_t^i = M^{2t} + 2\eta\sigma^2 \sum_{k=0}^{t-1} M^{2k} = M^{2m} + 2\eta\sigma^2 (I - M^2)^{-1}(I - M^{2t}).$$

Note that we used symmetry of $M$. □

Assume that $\eta(\lambda + M_x^2/n) < 1$. Then $M^t \to 0$ and

$$\mu_t^i \to B_i(I - M)^{-1} = \frac{1}{n} Y_i^T X \Sigma^{-1},$$

$$\Sigma_t^i \to 2\eta\sigma^2 (I - M^2)^{-1} = \sigma^2 (I - 2\eta\Sigma)^{-1}\Sigma^{-1} = \sigma^2 M^{-1}\Sigma^{-1}.$$

By Levy's continuity theorem, $V_t \to V_\infty \sim \otimes_{i=1}^n \mathcal{N}(Y_i^T X \Sigma^{-1}/n, \sigma^2 M^{-1}\Sigma^{-1})$ can be decomposed in independent rows which have the same covariance and different mean. Note that when $\eta \to 0$, we recover the Gibbs distribution.

## A.3. Proofs of Proposition 3.2, Proposition 3.3

In this section, we compute the trade-off functions $T(V_\infty, W_\infty)$ and $T(V_\infty z, W_\infty z)$. The following lemma characterizes the trade-off function between two Gaussian vectors with identical covariance and different mean. This function is expressed in terms $\| \cdot \|_\Sigma$ (Mahalanobis) norms, which we define below:

**Definition A.1** ($\| \cdot \|_\Sigma$ norm.)**.** Let $\mu \in \mathbb{R}^d$, $\Sigma \in \mathbb{R}^{d \times d}$ be an symmetric definite positive matrix. Then, the Mahalanobis norm is defined as:

$$\|\mu\|_\Sigma := \sqrt{\mu^T \Sigma^{-1} \mu}.$$

**Lemma A.1** (trade-off function between Gaussian vectors with different means.)**.** *Let $\mu, \mu' \in \mathbb{R}^d$, $\Sigma \in \mathbb{R}^{d \times d}$ be an symmetric definite positive matrix. Then,*

$$T(V, W) = G_{\|\mu - \mu'\|_\Sigma}.$$

*Let $\mu, \mu' \in \mathbb{R}^d$, $\Sigma \in \mathbb{R}^{d \times d}$ be a positive definite symmetric matrix. Let $V \sim \mathcal{N}(\mu, \Sigma)$, and $W \sim \mathcal{N}(\mu', \Sigma)$.*

*Proof.* Let $H_0 : \mathcal{N}(\mu, \Sigma)$, $H_1 : \mathcal{N}(\mu', \Sigma)$. We note the log likelihood ratio:

$$\begin{aligned} llk(x) &= (x - \mu)^T \Sigma^{-1}(x - \mu) - (x - \mu')^T \Sigma^{-1}(x - \mu') + C \\ &= (\mu' - \mu)^T \Sigma^{-1}(2x - (\mu + \mu')) + C \\ &= 2(\mu' - \mu)^T \Sigma^{-1} x - (\mu' - \mu)^T \Sigma^{-1}(\mu + \mu) + C. \end{aligned}$$

Following the Neyman-Pearson Lemma B.2, the most powerful test at level $\alpha$ has the form $T(x) = (\mu' - \mu)^T \Sigma^{-1} x > t_\alpha$.

Let $N \sim \mathcal{N}(0, I_d)$.

Under $H_0$:

$$(\mu' - \mu)^T \Sigma^{-1} V \sim \mathcal{N}((\mu' - \mu)^T \Sigma^{-1}\mu, (\mu' - \mu)^T \Sigma^{-1}(\mu' - \mu)) = \|\mu' - \mu\|_\Sigma N + (\mu' - \mu)^T \Sigma^{-1}\mu.$$

Then, the type I error is:

$$\alpha = P((\mu' - \mu)^T \Sigma^{-1} V > t_\alpha) = 1 - P\left(N \le \frac{t_\alpha - (\mu' - \mu)^T \Sigma^{-1}\mu}{\|\mu' - \mu\|_\Sigma}\right),$$

$$t_\alpha = (\mu' - \mu)^T \Sigma^{-1}\mu + \|\mu' - \mu\|_\Sigma \Phi^{-1}(1 - \alpha).$$

Under $H_1$:

$$(\mu' - \mu)^T \Sigma^{-1} V \sim \mathcal{N}((\mu' - \mu)^T \Sigma^{-1} \mu', (\mu' - \mu)^T \Sigma^{-1}(\mu' - \mu)) = \|\mu' - \mu\|_\Sigma N + (\mu' - \mu)^T \Sigma^{-1} \mu'.$$

Then, the type II error is given by:

$$
\begin{aligned}
\beta(\alpha) &= P((\mu' - \mu)^T \Sigma^{-1} x \le t_\alpha) = P\left(N \le \frac{t_\alpha - (\mu' - \mu)^T \Sigma^{-1} \mu'}{\|\mu' - \mu\|_\Sigma}\right) \\
&= P\left(N \le \frac{(\mu' - \mu)^T \Sigma^{-1} \mu + \|\mu' - \mu\|_\Sigma \Phi^{-1}(1 - \alpha) - (\mu' - \mu)^T \Sigma^{-1} \mu'}{\|\mu' - \mu\|_\Sigma}\right) \\
&= \Phi\left(\Phi^{-1}(1 - \alpha) + \frac{(\mu' - \mu)^T \Sigma^{-1}(\mu - \mu')}{\|\mu' - \mu\|_\Sigma}\right) \\
&= \Phi\left(\Phi^{-1}(1 - \alpha) - \|\mu' - \mu\|_\Sigma\right)
\end{aligned}
$$

Then, $T(V, W) = G_{\|\mu' - \mu\|_\Sigma}$. $\qquad\square$

Now we can prove Propositions 3.2, 3.3. We give a slightly more general result allowing $Y$ and $Y'$ to differ in multiple entries.

**Proposition A.2.** *Let $\Sigma = \frac{1}{n} X^T X + \lambda I$, $M = I - 2\eta\Sigma$ and denote by $A$ the square root of $\Sigma^{-1} M$. Assume that $Y$ and $Y'$ differ and that $\eta(\lambda + M_x^2/n) < 1$. Then:*

$$T(V_\infty, W_\infty) = G_{\|AX^T(Y - Y')\|/n\sigma}.$$

*Moreover, for two given datasets, the adversary can choose $z \in \mathbb{R}^d$ such that:*

$$T(V_\infty z, W_\infty z) = G_{|\sigma_{\max}(AX^T(Y - Y'))|/n\sigma}.$$

*In particular, when $Y'$ and $Y$ are adjacent (label DP), the adversary can choose $z \in \mathbb{R}^d$ such that:*

$$T(V_\infty z, W_\infty z) = T(V_\infty, W_\infty).$$

The first statement shows that the trade-off function can be improved from a sensivity $\|AX^T(Y - Y')\|/n\sigma$ to $|\sigma_{\max}(AX^T(Y - Y'))|/n\sigma$ when $Y$ and $Y'$ differ on multiple rows, but not when they are adjacent (as $Y - Y'$ is of rank 1).

We provide a unified proof below.

*Proof.* We note $B^2 = \Sigma^{-1} M^{-1}$. By Proposition A.1, $V_\infty$ and $W_\infty$ have independent rows and the $i$-th row of $V_\infty$ follows the distribution $(V_\infty)i \sim \mathcal{N}(Y_i^T X \Sigma^{-1}/n, \sigma^2 M^{-1} \Sigma^{-1})$. The square roots are defined because $\Sigma$ and $M$ commute. Using Lemma A.1, the trade-off function between $V_\infty$ and $W_\infty$ is $T(V_\infty, W_\infty) = G_{\sum_{i=1}^n \|\mu_i' - \mu_i\|_{\sigma^2 M^{-1} \Sigma^{-1}}}$, with $\mu' = \frac{1}{n} \Sigma^{-1} X^T Y'$, $\mu = \frac{1}{n} \Sigma^{-1} X^T y$. Then,

$$\|\mu_i' - \mu_i\|^2_{\sigma^2 M^{-1} \Sigma^{-1}} = (Y_i' - Y_i)^T X \Sigma^{-1} M X^T (Y_i' - Y_i)/\sigma^2 = \|AX^T(Y_i' - Y_i)\|^2/n^2\sigma^2.$$

Furthermore, for $z \in \mathbb{R}^d$, $V_\infty z \sim N((\Sigma^{-1} X^T Y_i \cdot z)_i/n, \sigma^2 z^T M^{-1} \Sigma^{-1} z I_n)$, Then,

$$T(V_\infty z, W_\infty z) = G_{\frac{\|z^T \Sigma^{-1} X^T(Y - Y')\|}{n\sigma \|Bz\|}}.$$

By noting the change of variable $u = Bz$ and using the invertibility of $B$, we get:

$$\sup_{z \ne 0} \frac{\|z^T \Sigma^{-1} X^T(Y - Y')\|}{\|Bz\|} = \sup_{u \ne 0} \frac{\|u^T(AX^T(Y - Y'))\|}{\|u\|} = \sigma_{\max}(AX^T(Y - Y')),$$

which corresponds to the 2-norm of $AX^T(Y - Y')$ and is obtained by setting $u^*$ the right singular vector corresponding to the largest singular value of $AX^T(Y - Y')$. In the setting of Label DP, $Y' - Y$ has rank 1, so $T(V_\infty z, W_\infty z) = T(V_\infty, W_\infty)$. $\qquad\square$

# B. Proofs of Section 4

## B.1. Proof of Lemma 4.1

We provide the characteristic functions of $VZ$ and $WZ$.

**Lemma 4.1.** *(Distribution of $VZ$ and $WZ$). The distribution $P_v$ has the following characteristic function:*

$$\phi_{P_v}(t) = \frac{\exp\left(\frac{-\sigma_z^2}{2}\operatorname{tr}(t^T vv^T t(I_l + \sigma_z^2\sigma_\theta^2 t^T t)^{-1})\right)}{\det\left(I_l + \sigma_z^2\sigma_\theta^2 t^T t\right)^{d/2}}.$$

*Proof.* Let $t \in \mathbb{R}^{n \times l}$. Assume that $\sigma_z = \sigma_\theta = 1$. We have:

$$\phi_{P_v}(t) = \mathbb{E}[\exp(i\operatorname{tr}(t^T(N+v)Z))] = \mathbb{E}[\mathbb{E}[\exp(i\operatorname{tr}(t^T(N+v)Z))|Z]].$$

Conditioned on $Z$, $(N+v)Z$ follows a matrix normal distribution $(N+v)Z|Z \sim \mathcal{MN}_{n,l}(vN, I_n, N^T N)$. Then,

$$\phi_{P_v}(t) = \mathbb{E}\left[\exp\left(i\operatorname{tr}(t^T vN) - \frac{1}{2}\operatorname{tr}(N^T Nt^T t)\right)\right].$$

Notice that $I_l + t^T t$ is symmetric positive definite. We note $A_t$ the square root of $I_l + t^T t$ and $B_t = iv^T tA_t^{-1}$. Then, leveraging the commutativity of trace operation and symmetry of $A_t$:

$$\phi_{P_v}(t) = (2\pi)^{-dl/2}\int \exp\left(i\operatorname{tr}(t^T vx) - \frac{1}{2}\operatorname{tr}(x^T xt^T t)\right)\exp\left(-\frac{1}{2}\operatorname{tr}(x^T x)\right)dx$$

$$= (2\pi)^{-dl/2}\int \exp\left(\frac{1}{2}\left(i\operatorname{tr}(x^T v^T t) + i\operatorname{tr}(t^T vx) - \operatorname{tr}(x^T xA_t^2)\right)\right)dx$$

$$= (2\pi)^{-dl/2}\int \exp\left(\frac{1}{2}\left(i\operatorname{tr}(v^T tx^T) + i\operatorname{tr}(xt^T v) - \operatorname{tr}(xA_t^2 x^T)\right)\right)dx$$

$$= (2\pi)^{-dl/2}\exp\left(\frac{1}{2}\operatorname{tr}(B_t B_t^T)\right)\int \exp\left(-\frac{1}{2}\left(\operatorname{tr}((xA_t - B_t)(xA_t - B_t)^T)\right)\right)dx$$

$$= (2\pi)^{-dl/2}\exp\left(\frac{1}{2}\operatorname{tr}(B_t B_t^T)\right)\int \exp\left(-\frac{1}{2}\left(\operatorname{tr}((x - B_t A_t^{-1})A_t^2(x - B_t A_t^{-1})^T)\right)\right)dx$$

$$= \exp\left(\frac{1}{2}\operatorname{tr}(B_t B_t^T)\right)\det\left(A_t^{-2}\right)^{d/2}$$

$$= \frac{\exp\left(\frac{-1}{2}\operatorname{tr}(v^T t(I_l + t^T t)^{-1}t^T v)\right)}{\det\left(I_l + t^T t\right)^{d/2}}.$$

Now, assume that $\sigma_z, \sigma_\theta \neq 1$. Then,

$$\phi_{P_v}(t) = \mathbb{E}[\exp(i\operatorname{tr}(t^T\sigma_z\sigma_\theta(N + v/\sigma_\theta)Z))]] = \frac{\exp\left(\frac{-\sigma_z^2}{2}\operatorname{tr}(t^T vv^T t(I_l + \sigma_z^2\sigma_\theta^2 t^T t)^{-1})\right)}{\det\left(I_l + \sigma_z^2\sigma_\theta^2 t^T t\right)^{d/2}}.$$

$\square$

## B.2. Proof of Lemma 4.2

We rewrite the Lemma 4.2 of approximation of trade-off functions and prove it.

**Lemma 4.2.** *(Approximate trade-off function). Let $P, Q, \tilde{P}, \tilde{Q}$ four distributions of $\mathbb{R}^d$. Let $\gamma = \max(TV(\tilde{P}, P), TV(\tilde{Q}, Q))$ where $TV$ denotes the total variation between distributions. Let $\alpha \in (\gamma, 1 - \gamma)$. Then,*

$$T(\tilde{P}, \tilde{Q})(\alpha + \gamma) - \gamma \leq T(P, Q)(\alpha) \leq T(\tilde{P}, \tilde{Q})(\alpha - \gamma) + \gamma.$$

*Proof.* Let $\alpha \in (0, 1 - \gamma)$. We define $0 \leq \Psi \leq 1$ as the most powerful test between $P$ and $Q$ at level $\alpha$. We have $T(P, Q)(\alpha) = 1 - \mathbb{E}_Q[\Psi]$.

Then, using the definition of TV distance and the fact that $0 \leq \Psi \leq 1$,

$$|\mathbb{E}_P[\Psi] - \mathbb{E}_{\tilde{P}}[\Psi]| \leq TV(\tilde{P}, P),$$

$$|\mathbb{E}_Q[\Psi] - \mathbb{E}_{\tilde{Q}}[\Psi]| \leq TV(\tilde{Q}, Q).$$

Then, using $\Psi$ is a test between $\tilde{P}$ and $\tilde{Q}$, $\Psi$ is at level $\alpha_{\tilde{P}, \tilde{Q}} \leq \alpha + TV(\tilde{P}, P)$ and is not necessarily optimal. This means that $T(\tilde{P}, \tilde{Q})(\alpha_{\tilde{P}, \tilde{Q}}) \leq 1 - \mathbb{E}_{\tilde{Q}}[\Psi]$. As $T(\tilde{P}, \tilde{Q})$ is non-increasing,

$$T(\tilde{P}, \tilde{Q})(\alpha + TV(\tilde{P}, P)) \leq T(\tilde{P}, \tilde{Q})(\alpha_{\tilde{P}, \tilde{Q}}) \leq 1 - \mathbb{E}_{\tilde{Q}}[\Psi] \leq 1 - \mathbb{E}_Q[\Psi] + TV(\tilde{Q}, Q) = T(P, Q)(\alpha) + TV(\tilde{Q}, Q).$$

As trade-off functions are non-increasing:

$$T(\tilde{P}, \tilde{Q})(\alpha + \gamma) - \gamma \leq T(P, Q)(\alpha).$$

For the other side of the inequality, take $\alpha' = \alpha + \gamma \in (\gamma, 1)$. We get:

$$T(\tilde{P}, \tilde{Q})(\alpha') \leq T(P, Q)(\alpha' - \gamma) + \gamma,$$

and leverage the symmetry of the setting, giving the result. $\qquad\square$

### B.3. Proof of Lemma 4.3

Before proving Lemma 4.3, we prove the following useful lemma:

**Lemma B.1.** *Les $\sigma > 0$. Let $N \in \mathbb{R}^{n \times d}, Z \in \mathbb{R}^{d \times l}$ be two independent standard Gaussian matrix. Let $v \in \mathbb{R}^{n \times d}$ be a deterministic matrix. Then, the distribution of $(\sigma N + v)Z$ is absolutely continuous with respect to the Lebesgue measure if and only if $d \geq \min\{n, l\}$.*

*Proof.* First, let us assume that $d < \min\{n, l\}$. We know that $\operatorname{rank}((\sigma N + v)Z) \leq \min\{\operatorname{rank}(N + v), \operatorname{rank}(Z)\}$, and because $N$ and $Z$ are Gaussian matrices, $\operatorname{rank}(N + v) = \min\{n, d\}$ and $\operatorname{rank}(Z) = \min\{d, l\}$. Then, $\operatorname{rank}((\sigma N + v)Z) \leq \min\{n, d, l\} \leq d$, and $(N + v)Z$ lies in the manifold $\mathcal{M}_{n,l}^d(\mathbb{R})$ of matrices of size $nl$ with rank $d$ which has dimension $d(n + l - d)$. Then, there exists a Borel set $A \subset \mathcal{M}_{n,l}^d(\mathbb{R})$ such that $P((\sigma N + v)Z \in A) > 0$. However, as $(n - d)(l - d) > 0$, $nl > d(n + l - d)$ and $\operatorname{Leb}_{nl}(A) = 0$. Thus, $(\sigma N + v)Z$ is not absolutely continuous with respect to the Lebesgue measure.

Now, assume that $d \geq n$. Let $A$ be Borel set of $\mathbb{R}^{n \times l}$ such that $\operatorname{Leb}_{nl}(A) = 0$. Then, we write $P((\sigma N + v)Z \in A) = \int P_{\sigma N + v}(x) P(xZ \in A) dx$. For all $x \in \mathbb{R}^{n \times d}$ such that $\operatorname{rank}(x) = n$, $xZ$ is a Gaussian matrix and admits a density [REF], which means that $P(xZ \in A) = 0$. Then, $P((\sigma N + v)Z \in A) = \int_{\operatorname{rank}(x) < n} P_{\sigma N + v}(x) P(xZ \in A) dx$. Also, for all $x \in \mathbb{R}^{n \times d}$, $P(xZ \in A) \leq 1$. Then, $P((\sigma N + v)Z \in A) \leq \int_{\operatorname{rank}(x) < n} P_{\sigma N + v}(x) dx = P(\operatorname{rank}(\sigma N + v) < n) = 0$, because $N + v$ is a Gaussian matrix and is absolutely continuous with respect to the Lebesgue measure. Then, $(N + v)Z$ is absolutely continuous with respect to the Lebesgue measure. In the case $n \geq l$, we apply the same reasoning to the decomposition $P((\sigma N + v)Z \in A) = \int P_Z(y) P((\sigma N + v)y \in A) dy$. $\qquad\square$

**Lemma 4.3.** *Let $G \sim \mathcal{N}(0, 1)$. Then, there exists $A_{\|v\|} > 0$ such that:*

$$TV\left(\sqrt{d}(\sigma_\theta N + v)Z, \sigma_z \sqrt{d\sigma_\theta^2 + \|v\|^2} G\right) \leq \frac{A_{\|v\|}}{d}.$$

*Proof.* We note $U \in \mathbb{R}^d$ the orthogonal matrix such that $vU^T = \bar{v} = \frac{1}{\sqrt{d}}(\|v\|, \dots, \|v\|)$. Then, using invariance of Gaussian matrices by orthogonal transformation, $VZ = (N + \bar{v}U^T)Z = (NU + \bar{v})U^T Z \stackrel{d}{=} (N + \bar{v})Z$. We write $VZ = \sum_{k=1}^d V_k Z_k \stackrel{d}{=} \sum_{k=1}^d (\sigma_\theta N_k + \frac{1}{\sqrt{d}}\|v\|)Z_k$, which is a sum of iid components. By Lemma B.1, $(\sigma_\theta N_1 + \frac{1}{\sqrt{d}}\|v\|)Z_1$

is absolutely continuous with respect to the Lebesgue measure. We compute:

$$\mathbb{E}\left[(\sigma_\theta N_1 + \frac{1}{\sqrt{d}}\|v\|)Z_1\right] = \mathbb{E}\left[(\sigma_\theta N_1 + \frac{1}{\sqrt{d}}\|v\|)\right]\mathbb{E}[Z_1] = 0,$$

$$\mathbb{E}\left[(\sigma_\theta N_1 + \frac{1}{\sqrt{d}}\|v\|)^2 Z_1^2\right] = \mathbb{E}\left[(\sigma_\theta N_1 + \frac{1}{\sqrt{d}}\|v\|)^2\right]\mathbb{E}[Z_1^2] = (\sigma_\theta^2 + \frac{1}{d}\|v\|^2)\sigma_z^2,$$

$$\mathbb{E}\left[(\sigma_\theta N_1 + \frac{1}{\sqrt{d}}\|v\|)^2 Z_1^3\right] = 0.$$

$$= \mathbb{E}\left[\left(\sigma_\theta N_1 + \frac{1}{\sqrt{d}}\|v\|\right)^4 Z_1^4\right] = 3\sigma_z^4(3\sigma_\theta^4 + 6\sigma_\theta^2\|v\|^2/d + \|v\|^4/d^2)$$

Then, $\frac{1}{\sigma_z\sqrt{\sigma_\theta^2 + \frac{1}{d}\|v\|^2}}(\sigma_\theta N_1 + \frac{1}{\sqrt{d}}\|v\|)Z_1$ match the 3 first moments of a Gaussian variable.

Then, leveraging Theorem 4.2,

$$TV\left(VZ, \sigma_z\sqrt{\sigma_\theta^2 + \frac{1}{d}\|v\|^2}G\right) \le C\left(1 + \frac{\mathbb{E}\left[\left(\sigma_\theta N_1 + \frac{1}{\sqrt{d}}\|v\|\right)^4 Z_1^4\right]}{\sigma_z^4\left(\sqrt{\sigma_\theta^2 + \frac{1}{d}\|v\|^2}\right)^4}\right)\frac{1}{d} \le \frac{A_{\|v\|}}{d},$$

with $A_{\|v\|} = C(9 - \frac{6}{(1+d\sigma_\theta^2/\|v\|^2)^2})$. $\qquad\square$

This means that for any $v \in \mathbb{R}^d$, $A_{\|v\|} \le 9C$.

Note that $C$ does not depend on $\sigma_\theta, v$ and $d$. In fact, noting $Y_k = \frac{1}{\sigma_z\sqrt{\sigma_\theta^2 + \frac{1}{d}\|v\|^2}}(\sigma_\theta N_k + \frac{1}{\sqrt{d}}\|v\|)Z_k$, we have $\mathrm{Cov}(Y_1) = I_1 = 1$, and we apply Theorem 4.2 on the random variable $\frac{1}{\sqrt{d}}\sum_{k=1}^d Y_k$, removing the dependence. The final bound is obtained by invariance of TV distance by invertible transformation.

### B.4. Proof of Proposition 4.1

Before proving the result, we recall the Neyman-Pearson lemma.

**Lemma B.2** (Neyman-Pearson lemma (Lehmann and Romano, 2006)). *Let $P$ and $Q$ be probability distributions on $\Omega$ admitting densities $p$ and $q$, respectively with respect to some measure $\nu$. For the hypothesis testing problem $H_0 : P$ vs $H_1 : Q$, a test $\phi : \Omega \to [0, 1]$ is the most powerful test at level $\alpha$ if and only if there exists a constant $h > 0$ such that $\phi$ has the form:*

$$\phi(w) = \begin{cases} 1 \ \text{if } q(w) > hp(w) \\ 0 \ \text{if } q(w) < hp(w), \end{cases}$$

*and $\mathbb{E}_P[\phi] = \alpha$.*

**Proposition 4.1.** (Trade-off function between Gaussians with different variance). *Let $\sigma_1, \sigma_2 > 0$. Then,*

$$T(\mathcal{N}(0, \sigma_1^2), \mathcal{N}(0, \sigma_2^2)) = \begin{cases} T_1(\alpha) \ \text{if } \sigma_1 \le \sigma_2, \\ T_2(\alpha) \ \text{else,} \end{cases}$$

$$\text{where} \quad T_1(\alpha) = 2\Phi\left(\frac{\sigma_1}{\sigma_2}\Phi^{-1}(1 - \alpha/2)\right) - 1,$$

$$T_2(\alpha) = 2 - 2\Phi\left(\frac{\sigma_1}{\sigma_2}\Phi^{-1}((\alpha + 1)/2)\right).$$

*We denote this trade-off function by $\tilde{G}_{(\sigma_1, \sigma_2)}$.*

*Proof.* Assume that $0 \leq \sigma_1 \leq \sigma_2$. We note $H_0 : X \sim \mathcal{N}(0, \sigma_1^2) = P$, $H_1 : X \sim \mathcal{N}(0, \sigma_2^2) = Q$. We write the log-likelihood ratio between $P$ and $Q$:

$$llk(x) = \frac{x^2}{2}\left(\frac{1}{\sigma_2^2} - \frac{1}{\sigma_1^2}\right).$$

Noting that $\left(\frac{1}{\sigma_2^2} - \frac{1}{\sigma_1^2}\right) \geq 0$ and using the Neyman Pearson Lemma B.2, the most powerful test at level $\alpha$ has the form $T(x) = x^2 \geq t_\alpha$. Let $N \sim \mathcal{N}(0,1)$. The type I error is:

$$\alpha = P(\sigma_1 N^2 > t_\alpha) = 2(1 - P(0 \leq N \leq t_\alpha/\sigma_1)) = 2(1 - \Phi(t_\alpha/\sigma_1)).$$

Then, $t_\alpha = \sigma_1 \Phi^{-1}(1 - \alpha/2)$. Also, the type II error writes:

$$\beta(\alpha) = P(\sigma_2 N^2 \leq t_\alpha) = 2\Phi(t_\alpha/\sigma_2) - 1 = 2\Phi\left(\frac{\sigma_1}{\sigma_2}\Phi^{-1}(1 - \alpha/2)\right) - 1.$$

Also, if $\sigma_1 > \sigma_2$, the log-likelihood ratio $llk(x) \leq 0$. Then, the most powerful test at level $\alpha$ has the form $T(x) = x^2 \leq t_\alpha$. The type I error is:

$$\alpha = P(\sigma_1 N^2 \leq t_\alpha) = 2P(0 \leq N \leq t_\alpha/\sigma_1) = 2\Phi(t_\alpha/\sigma_1) - 1.$$

Then, $t_\alpha = \sigma_1 \Phi^{-1}\left(\frac{1+\alpha}{2}\right)$. Also, the type II error writes:

$$\beta(\alpha) = P(\sigma_2 N^2 > t_\alpha) = 2(1 - \Phi(t_\alpha/\sigma_2)) = 2\left(1 - \Phi\left(\frac{\sigma_1}{\sigma_2}\Phi^{-1}\left(\frac{1+\alpha}{2}\right)\right)\right).$$

Combining both tests gives the trade-off function. $\qquad\square$

We also give the trade-off function between Gaussian with different variances. This trade-off function depends on the cdf and the inverse cdf of weighted chi squared variable, which are not trivial to compute in practice.

**Lemma B.3** (Trade-off function between Gaussians matrices with different variances). *Let $v, w \in \mathbb{R}^{n \times d}$. Let $\Sigma_v = \sigma^2 I_n + vv^T \in \mathbb{R}^{n \times n}$, $\Sigma_w = \sigma^2 I_n + ww^T \in \mathbb{R}^{n \times n}$, $P = \mathcal{N}(0, \Sigma_v \otimes I_l)$ and $Q = \mathcal{N}(0, \Sigma_w \otimes I_l)$. We note $\lambda_1, \ldots, \lambda_n$ the eigenvalues of $\Sigma_v^{-1/2}\Sigma_w\Sigma_v^{-1/2}$. We note $F_{\lambda_1,\ldots,\lambda_n}(l, x)$ the distribution of the weighted sum of $n$ independent $\chi_l^2$ variables with weights $\lambda_1, \ldots, \lambda_n$ at $x$. Then,*

$$T(P,Q)(\alpha) = F_{(\lambda_k - 1; k \in [\![1,n]\!])}\left(l, F^{-1}_{(1 - 1/\lambda_k; k \in [\![1,n]\!])}(l, 1 - \alpha)\right).$$

*Proof.* We start with the case $l = 1$. $vv^T$ is positive semi-definite, so $\Sigma_v$ is positive definite and $\Sigma^{-1}$ exists. The log likelihood ratio between $P$ and $Q$ is:

$$llk(x) = x^T(\Sigma_v^{-1} - \Sigma_w^{-1})x + C.$$

Using the Neyman Pearson Lemma B.2, the most powerful test at level $\alpha$ has the form $T(x) = x^T(\Sigma_v^{-1} - \Sigma_w^{-1})x = x^T\Sigma_v^{-1/2}(I_n - \Sigma_v^{1/2}\Sigma_w^{-1}\Sigma_v^{1/2})\Sigma_v^{-1/2}x \geq t_\alpha$. We investigate the law of $T$ under $P$ and $Q$. We note $M = \Sigma_v^{-1/2}\Sigma_w\Sigma_v^{-1/2}$. $M$ is symmetric and positive definite. Then, we note $M = U^T DU$, with $D = diag(\lambda_1, \ldots, \lambda_n)$.

Let $N \sim \mathcal{N}(0, I_d)$.

Under $P$: $X = \Sigma_v^{1/2}N$ and $\Sigma_v^{-1/2}X = N$:

$$T(X) = N^T(I_n - M^{-1})N = N^T(I_n - D^{-1})N = \sum_{k=1}^n (1 - 1/\lambda_k)N_k^2,$$

Under $Q$: $X = \Sigma_w^{1/2}N$ and $U\Sigma_v^{-1/2}X \sim \mathcal{N}(0, D)$. Then, writing $T(X) = X^T\Sigma_v^{-1/2}U^T(I_n - D^{-1})U\Sigma_v^{-1/2}X$:

$$T(X) = N^T D^{1/2}(I_n - D^{-1})D^{1/2}N = N^T(D - I_n)N = \sum_{k=1}^n (\lambda_k - 1)N_k^2,$$

Then, the type I error is:

$$P\left(\sum_{k=1}^{n}(1-1/\lambda_k)N_k^2 > t_\alpha\right) = \alpha \implies t_\alpha = F_{(1-1/\lambda_k;k\in[\![1,n]\!])}^{-1}(1, 1-\alpha).$$

Finally, the type II error is given by:

$$\beta(\alpha) = P\left(\sum_{k=1}^{n}(\lambda_k-1)N_k^2 \leq t_\alpha\right) = F_{(\lambda_k-1;k\in[\![1,n]\!])}\left(F_{(1-1/\lambda_k;k\in[\![1,n]\!])}^{-1}(1, 1-\alpha)\right).$$

In the case $l > 1$, the eigenvalues of $M_l = (\Sigma_v \otimes I_l)^{-1/2}(\Sigma_w \otimes I_l)(\Sigma_v \otimes I_l)^{-1/2} = M \otimes I_l$ are $\underbrace{\lambda_1, \ldots, \lambda_1}_{l \text{ times}}, \ldots, \underbrace{\lambda_n, \ldots, \lambda_n}_{l \text{ times}}$.

The log likelihood ratio between $P$ and $Q$ is:

$$llk(x) = x^T((\Sigma_v \otimes I_l)^{-1} - (\Sigma_w \otimes I_l)^{-1})x + C = x^T((\Sigma_v^{-1} - \Sigma_w^{-1}) \otimes I_l)x + C.$$

As in the case $l = 1$, we can write the distribution of the test $T = x^T((\Sigma_v^{-1} - \Sigma_w^{-1}) \otimes I_l)x$ under $P$ and $Q$ and compute the type I and II errors, giving the desired result. Let $N \sim \mathcal{N}(0, I_n \otimes I_l)$.

In the case $\lambda_1 = \cdots = \lambda_n = \lambda > 1$, the type I error is:

$$P\left(\sum_{k'=1}^{m}\sum_{k=1}^{n}(1-1/\lambda_k)N_{k,k'}^2 > t_\alpha\right) = \alpha \implies P((1-1/\lambda)\chi_{nl} > t_\alpha) = \alpha \implies t_\alpha = (1-1/\lambda)\Phi_{\chi_{nl}^2}^{-1}(1-\alpha),$$

where $\chi_{nl}^2$ is the cdf of a chi-squared variable with $nl$ degrees of freedom. The type II error is given by:

$$\beta(\alpha) = P\left(\sum_{k'=1}^{m}\sum_{k=1}^{n}(\lambda_k-1)N_{k,k'}^2 \leq t_\alpha\right) = P((\lambda-1)\chi_{nl}^2 \leq t_\alpha) = \Phi_{\chi_{nl}^2}\left(\frac{1}{\lambda}\Phi_{\chi_{nl}^2}^{-1}(1-\alpha)\right).$$

$\square$

### B.5. Proof of Theorem 4.3

**Theorem 4.3.** *Let $d > 0$. Then, there exists a universal constant $C > 0$ such that for all $\alpha \in (C/d, 1-C/d)$:*

$$\tilde{G}_{\Lambda(\sigma_\theta, d, \|v\|, \|w\|)}\left(\alpha + \frac{C}{d}\right) - \frac{C}{d} \leq T(VZ, WZ)(\alpha),$$

$$\tilde{G}_{\Lambda(\sigma_\theta, d, \|v\|, \|w\|)}\left(\alpha - \frac{C}{d}\right) + \frac{C}{d} \geq T(VZ, WZ)(\alpha).$$

*Proof.* Let $d > 0$. Based on Lemma 4.3, there exists $C > 0$ such that:

$$TV\left(VZ, \sigma_z\sqrt{\sigma_\theta^2 + \frac{1}{d}\|v\|^2}G\right) \leq \frac{C}{d}\left(9 - \frac{6}{(1 + d\sigma_\theta^2/\|v\|^2)^2}\right) \leq \frac{9C}{d},$$

$$TV\left(WZ, \sigma_z\sqrt{\sigma_\theta^2 + \frac{1}{d}\|w\|^2}G\right) \leq \frac{C}{d}\left(9 - \frac{6}{(1 + d\sigma_\theta^2/\|w\|^2)^2}\right) \leq \frac{9C}{d}.$$

Then, leveraging Lemma 4.2, for $\alpha \in (C/d, 1-C/d)$:

$$T(VZ, WZ)(\alpha) \geq T\left(\sigma_z\sqrt{\sigma_\theta^2 + \frac{1}{d}\|v\|^2}G, \sigma_z\sqrt{\sigma_\theta^2 + \frac{1}{d}\|w\|^2}G\right)\left(\alpha + \frac{C}{d}\right) - \frac{9C}{d},$$

$$T(VZ, WZ)(\alpha) \leq T\left(\sigma_z\sqrt{\sigma_\theta^2 + \frac{1}{d}\|v\|^2}G, \sigma_z\sqrt{\sigma_\theta^2 + \frac{1}{d}\|w\|^2}G\right)\left(\alpha - \frac{C}{d}\right) + \frac{9C}{d},$$

giving the desired result. $\square$

## B.6. Proof of Theorem 4.5

**Theorem 4.5.** (Convergence of product of Gaussian matrices, shifted version). *Let $N \in \mathbb{R}^{n \times d}, Z, Z' \in \mathbb{R}^{d \times l}, G \in \mathbb{R}^{n \times l}$ be independent standard Gaussian matrices. Let $s = \text{rank}(v)$. Assume that $d \geq \max\{n, l\}$. Then, there exists $C' > 0$ such that:*

$$TV\left((\sigma_\theta N + v)Z, \sigma_\theta \sqrt{d-s}G + vZ'\right) \leq C'\sqrt{\frac{nls}{d-s}}.$$

*Proof.* We assume that $d \geq \max\{n, l, s\}$. Let $N \in \mathbb{R}^{n \times d}, Z, Z' \in \mathbb{R}^{d \times l}, G \in \mathbb{R}^{n \times l}$ be independent standard Gaussian matrices. We note the singular value decomposition of $v = F\Sigma S^T$. Then, $(\sigma_\theta N + v)Z = (\sigma_\theta N + F\Sigma S^T)Z = F(\sigma_\theta F^T NS + \Sigma)S^T Z \overset{d}{=} F(\sigma_\theta N + \Sigma)Z$, by invariance of Gaussian matrices through orthogonal transformations. Because $F$ is invertible, we can write: $TV((\sigma_\theta N + v)Z, \sigma_\theta \sqrt{d-s}G + vZ') = TV(F(\sigma_\theta N + \Sigma)Z, \sigma_\theta \sqrt{d-s}G + vZ') = TV((\sigma_\theta N + \Sigma)Z, \sigma_\theta \sqrt{d-s}G + \Sigma Z')$.

We observe that $NZ$ is decomposed into a sum of independent bits: $NZ = \sum_{k=1}^{d} N_{\cdot,k}Z_k$. Then,

$$
\begin{aligned}
&TV((\sigma_\theta N + \Sigma)Z, \sigma_\theta \sqrt{d-s}G + \Sigma Z') \\
&= TV\left(\sum_{k=s+1}^{d} N_{\cdot,k}Z_k + \sum_{k=1}^{s}(N_{\cdot,k} + \Sigma_{\cdot,k})Z_k, \sigma_\theta \sqrt{d-s}G + vZ'\right) \\
&\leq TV\left(\sum_{k=s+1}^{d} N_{\cdot,k}Z_k + \sum_{k=1}^{s}(N_{\cdot,k} + \Sigma_{\cdot,k})Z_k, \sigma_\theta \sqrt{d-s}G + \sum_{k=1}^{s}(N_{\cdot,k} + \Sigma_{\cdot,k})Z_k\right) \quad (1) \\
&\quad + TV\left(\sigma_\theta \sqrt{d-s}G + \sum_{k=1}^{s}(N_{\cdot,k} + \Sigma_{\cdot,k})Z_k, \sigma_\theta \sqrt{d-s}G + \Sigma Z'\right). \quad\quad (2)
\end{aligned}
$$

Using the data-processing inequality for TV distance for the map $(x, y) \mapsto x + y$,

$$
\begin{aligned}
(1) &\leq TV\left(\left(\sigma_\theta \sum_{k=s+1}^{d} N_{\cdot,k}M_k, \sum_{k=1}^{s}(\sigma_\theta N_{\cdot,k} + \Sigma_{\cdot,k})Z_k\right), \left(\sigma_\theta \sqrt{d-s}G, \sum_{k=1}^{s}(\sigma_\theta N_{\cdot,k} + \Sigma_{\cdot,k})Z_k\right)\right) \\
&= TV\left(\sigma_\theta \sum_{k=s+1}^{d} N_{\cdot,k}M_k, \sigma_\theta \sqrt{d-s}G\right) \leq C\sqrt{\frac{nl}{d-s}},
\end{aligned}
$$

by independence of the components in the pair and using Theorem 4.5 for a product or size $d - s$. Also, using Pinsker inequality:

$$(2) \leq \sqrt{\frac{1}{2}D_{KL}\left(\sigma_\theta \sqrt{d-s}G + \Sigma Z', \sigma_\theta \sqrt{d-s}G + \sum_{k=1}^{s}(\sigma_\theta N_{\cdot,k} + \Sigma_{\cdot,k})Z_k\right)}.$$

Because of independence of components in the sum, this divergence corresponds to the KL divergence between the convolution of two distributions by a third distribution. We find an upper bound to this divergence using the chain rule for KL divergence and a technique proof similar to the shift-reduction lemma (Feldman et al., 2018). Shift reduction lemma allows to upper bound the divergence between convolution of two distributions by the same distribution by a coupling technique and the use of chain rule.

**Lemma B.4** (Shift-reduction for KL divergences). *Let X,Y and N be three independent random vectors. Then, for any random variable W that depend on X,*

$$D_{KL}(X + N, Y + N) \leq D_{KL}(X + W, Y) + \mathbb{E}_{w \sim W}[D_{KL}(N - w, N)].$$

*Proof.* Let $X, Y$ and $N$ be three independent random vectors. Let $W$ be a random variable. Then, we observe that $X + N = X + W - W + N$. We apply the post-processing inequality for KL divergence under the map $f : (x, y) \mapsto x + y$:

$$D_{KL}(X + N, Y + N) \leq D_{KL}((X + W, -W + N)(Y, N)).$$

Using the independence between $N$ and the other variables, we obtain:

$$D_{KL}(X + N, Y + N) = D_{KL}(X + W, Y) + \mathbb{E}_{y \sim p_{X+W}}[D_{KL}(N - W | X + W = y, N)].$$

Then, using convexity of KL divergences and writing $P_{N-W|X+W=x}(y) = \int P_N(y + w) P_{W|X+W=x}(w) dw$, we obtain

$$D_{KL}(X + N, Y + N) \leq D_{KL}(X + W, Y) + \mathbb{E}_{y \sim P_{X+W}}[\mathbb{E}_{w \sim W|X+W=y}[D_{KL}(N - w, N)] = \mathbb{E}_{w \sim W}[D_{KL}(N - w, N)].$$

$\square$

Then, for $\alpha > 1$, setting a shift $W = \sigma_\theta \sum_{k=1}^{s} N_{\cdot,k} Z'_k$, we have in distribution, $\Sigma Z' + W \overset{d}{=} \sum_{k=1}^{s} (\sigma_\theta N_{\cdot,k} + \Sigma_{\cdot,k}) Z_k$. It means that:

$$D_{KL}\left(\Sigma Z' + W, \sum_{k=1}^{s}(\sigma_\theta N_{\cdot,k} + \Sigma_{\cdot,k}) Z_k\right) = 0.$$

Also, $D_{KL}(\sigma_\theta \sqrt{d-s} G - w, \sigma_\theta \sqrt{d-s} G) = \frac{\|w\|^2}{2\sigma_\theta^2(d-s)}$.

Using the shift-reduction lemma, we get:

$$D_{KL}\left(\Sigma Z' + \sigma_\theta \sqrt{d-s} G, \sum_{k=1}^{s}(N_{\cdot,k} + \Sigma_{\cdot,k}) Z_k + \sigma_\theta \sqrt{d-s} G\right) \leq \mathbb{E}_{w \sim W}[D_{KL}(\sigma_\theta \sqrt{d-s} G - w, \sigma_\theta \sqrt{d-s} G)]$$

$$\leq \mathbb{E}_{w \sim W}\left[\frac{\|w\|^2}{2\sigma_\theta^2(d-s)}\right]$$

$$\leq \frac{nls}{2(d-s)},$$

where expectation is obtained using independence between $N$ and $Z$,

$$\mathbb{E}[\|W\|^2] = \sigma_\theta^2 \sum_{i=1}^{n} \sum_{j=1}^{l} \mathbb{E}\left[\left(\sum_{k=1}^{s} N_{i,k} Z_{k,j}\right)^2\right] = \sigma_\theta^2 nl \sum_{k=1}^{s} \sum_{k'=1}^{s} \mathbb{E}[N_{i,k} N_{i,k'}] \mathbb{E}[Z_{k,j} Z_{k',j}] = \sigma_\theta^2 nls.$$

Finally, we find:

$$TV((\sigma_\theta N + v) Z, \sigma_\theta \sqrt{d-s} G + v Z') \leq C\sqrt{\frac{nl}{d-s}} + \sqrt{\frac{nls}{4(d-s)}} \leq C'\sqrt{\frac{nls}{d-s}}.$$

We obtain the result of the lemma by applying invariance of TV distance under rescaling.

$\square$

### B.7. Experiments and Interpretation with Rényi divergences

In this section, we explain the setup of our experiments, and compute upper bounds for Rényi divergences between Gaussian matrices with different covariance structures.

**Rényi divergence between Gaussian variables.**

Let $s > 0$. The Rényi divergence between $\mathcal{N}(0, s + v^2)$ and $\mathcal{N}(0, s + w^2)$ is maximized by setting $v = r_{s,\Delta}, w = v + \Delta$, for some value $0 < r_{s,\Delta} < 1$.

**Lemma B.5** (maximum of Rényi divergence between Gaussian variables with same mean and different variance).

$$\sup_{|v-w|\leq\Delta} D_\alpha(\mathcal{N}(0, s + v^2), \mathcal{N}(0, s + w^2)) = \begin{cases} \frac{1}{2(\alpha-1)}(\alpha\log(r_{s,\Delta}) - \log(\alpha r_{s,\Delta} + 1 - \alpha)) \text{ if } s \geq \alpha(\alpha-1)\Delta^2, \\ +\infty \text{ else,} \end{cases}$$

*where* $r_{s,\Delta} = \dfrac{2s + \Delta^2 - \Delta\sqrt{\Delta^2 + 4s}}{2s}$.

*Proof.* For $r > 0$, we note $f(r) = \alpha\log(r) - \log(\alpha r + 1 - \alpha)$, and we note $r(v, w) = \frac{s+v^2}{s+w^2}$.

The Rényi divergence of order $\alpha > 1$ between $(\mathcal{N}(0, s + v^2)$ and $\mathcal{N}(0, s + w^2)$ is given by $D_\alpha(\mathcal{N}(0, s+v^2), \mathcal{N}(0, s+w^2)) = \frac{1}{2(\alpha-1)}f(r(v, w))$. We note $r_{\max} = \sup_{|v-w|\leq\Delta} r(v, w)$, $r_{\min} = \sup_{|v-w|\leq\Delta} r(v, w)$. We study the function $f$:

$$f'(r) = \alpha\left(\frac{1}{r} - \frac{1}{\alpha r + 1 - \alpha}\right).$$

$f'(r) \leq 0$ if $r \in (1 - 1/\alpha, 1)$ and else $f'(r) \geq 0$. This means that $\sup_{|v-w|\leq\Delta} f(r(v, w)) = \max\{f(r_{\max}), f(r_{min})\}$.

Now, we compute $r_{\max}$ and $r_{\min}$. We first observe that $r_{\max} = 1/r_{\min}$. Writing $v = w + x$, and $|x| \leq \Delta$, we have $r(w + x, w) = \frac{s+(w+x)^2}{s+w^2}$. Also, $r_{\max}$ is obtained for $v = w + \Delta$, and $w$ positive. In fact, if $w < 0$, for $x \in \mathbb{R}$ such that $|x| \leq \Delta$, $r(w + x, w) = r(-w - x, w) \leq r(-w + \Delta, w)$. We derivate: $\frac{d}{dw}r(w + \Delta, w) = 2\Delta\frac{s-w\Delta-w^2}{(s+w^2)^2}$, which has two roots. We note $w_{\max} = \frac{1}{2}(\sqrt{\Delta^2 + 4s} - \Delta)$, which verifies $w_{\max} + \Delta = s/w_{\max}$. Then,

$$r_{\max} = \frac{s + (w_{\max} + \Delta)^2}{s + w_{\max}^2} = \frac{s + (s/w_{\max})^2}{s + w_{\max}^2} = \frac{s}{w_{\max}^2} = \frac{2s}{2s + \Delta^2 - \Delta\sqrt{\Delta^2 + 4s}}.$$

Then, we investigate $\max\{f(x), f(1/x)\}$ for $x \in (1, \alpha/(\alpha-1))$:

Let, for $x \in (1, \alpha/(\alpha-1))$, $g(x) = x^{1-2\alpha} - \frac{\alpha+(1-\alpha)x}{\alpha x+1-\alpha}$. Then,

$$g'(x) = (1 - 2\alpha)x^{-2\alpha} - \frac{(1-\alpha)(\alpha x + 1 - \alpha) - \alpha(\alpha + (1-\alpha)x)}{(\alpha x + (1-\alpha))^2} = (1 - 2\alpha)\left(x^{-2\alpha} - \frac{1}{(\alpha + (1-\alpha)x)^2}\right).$$

Given that $\alpha > 1$, $x \mapsto x^\alpha$ is convex on $(1, \alpha/(\alpha-1))$, $x^\alpha \geq 1 + \alpha x > \alpha x + 1 - \alpha$, and, by monotonicity, $x^{-2\alpha} \leq \frac{1}{(\alpha+(1-\alpha)x)^2}$. Then, $g'(x) \geq 0$.

Also, $g(0) = 0$. Therefore, for $x \in (1, \alpha/(\alpha-1)), g(x) \geq 0$. By monotonicity of the logarithm,

$$(1 - 2\alpha)\log(x) \geq \log\left(\frac{\alpha + (1-\alpha)x}{\alpha x + 1 - \alpha}\right) \implies f(1/x) \geq f(x).$$

Then, the supremum of the Rényi divergence attained for $r = r_{\min}$ and is given by:

$$\sup_{|v-w|\leq\Delta} D_\alpha(\mathcal{N}(0, s + v^2), \mathcal{N}(0, s + w^2)) = \begin{cases} \frac{1}{2(\alpha-1)}(\alpha\log(r_{\min}) - \log(\alpha r_{\min} + 1 - \alpha)) \text{ if } s \geq \alpha(\alpha-1)\Delta^2, \\ +\infty \text{ else,} \end{cases}$$

*where* $r_{\min} = \dfrac{2s + \Delta^2 - \Delta\sqrt{\Delta^2 + 4s}}{2s}$.

$\square$

Performing an asymptotic expansion, we recover convergence rates for the Rényi divergence:

**Proposition B.1** (Rényi divergence rate of convergence – univariate case).

$$\sup_{|v-w|\leq\Delta} D_\alpha(\mathcal{N}(0, \sigma_\theta^2 d + v^2), \mathcal{N}(0, \sigma_\theta^2 d + w^2)) = \frac{\alpha\Delta^2}{4d\sigma_\theta^2} + o(d^{-1}).$$

*Proof.* Note $s = \sigma^2 d$. We first compute an asymptotical expansion of $r_{s,\Delta}$.

$$r_{s,\Delta} = \frac{2s + \Delta^2 - \Delta\sqrt{\Delta^2 + 4s}}{2s} = 1 + \frac{\Delta^2}{2s} - \frac{\Delta}{\sqrt{s}}\sqrt{\frac{\Delta^2}{4s} + 1} = 1 + \frac{\Delta^2}{2s} - \frac{\Delta}{\sqrt{s}} + o(1).$$

Also, we have $\alpha \log(1 + r) - \log(1 + \alpha r) = \alpha(r - r^2/2 + o(r^2)) - (\alpha r - (\alpha r)^2/2 + o(r^2)) = \frac{\alpha(\alpha-1)}{2}r^2 + o(r^2)$. Combining both expansions,

$$\frac{1}{2(\alpha-1)}(\alpha \log(r_{s,\Delta}) - \log(\alpha r_{s,\Delta} + 1 - \alpha)) = \frac{\alpha\Delta^2}{4d} + o(d^{-1}).$$

$\square$

### Rényi divergence between Gaussian matrices.

Now, we compute an upper bound of $\sup_{\|v-w\| \leq \Delta} D_\alpha(\sqrt{s}G + vZ, \sqrt{s}G + wZ)$.

**Lemma B.6** (Upper bound of Rényi divergence between Gaussian variables with same mean and different variance)**.** *Let* $G \in \mathbb{R}^{n \times l}, G \in \mathbb{R}^{d \times l}$ *be Gaussian matrices, and* $v, w \in \mathbb{R}^{n \times d}$. *Then,*

$$\sup_{\|v-w\| \leq \Delta} D_\alpha(\sqrt{s}G + vZ, \sqrt{s}G + wZ) \leq \begin{cases} \frac{nl}{2(\alpha-1)}(\alpha \log(r_{s,\Delta}) - \log(\alpha r_{s,\Delta} + 1 - \alpha)) \text{ if } s \geq \alpha(\alpha-1)\Delta^2, \\ +\infty \text{ else,} \end{cases}$$

*where* $r_{s,\Delta} = \frac{2s + \Delta^2 - \Delta\sqrt{\Delta^2 + 4s}}{2s}$.

*Proof.* Because $vZ$ is composed of independent columns, we can write $D_\alpha(\sqrt{s}G + vZ, \sqrt{s}G + wZ) = lD_\alpha(\sqrt{s}G_1 + vZ_1, \sqrt{s}_1 + wZ_1)$. Noting $\Sigma_v = sI_n + vv^T$, $\Sigma_w = sI_n + ww^T$, we reduce to the problem:

$$D_\alpha(sG + vZ, sG + wZ) = lD_\alpha(\mathcal{N}(0, \Sigma_v), \mathcal{N}(0, \Sigma_w)).$$

$\Sigma_v$ is positive definite and admit a square root. We note $M_{v,w} = \Sigma_v^{-1/2}\Sigma_w\Sigma_v^{-1/2}$, and note $\lambda_1, \ldots, \lambda_n$ its eigenvalues. Then, by invariance of Rényi divergence my invertible matrix multiplication, the Rényi divergence can be written:

$$D_\alpha(\mathcal{N}(0, \Sigma_v), \mathcal{N}(0, \Sigma_w)) = D_\alpha(\mathcal{N}(0, I_n), \mathcal{N}(0, M_{v,w})) = \frac{1}{2(\alpha-1)}(\alpha \log(\det(M_{v,w}) - \log(\alpha M_{v,w} + 1 - \alpha))$$

$$= \frac{1}{2(\alpha-1)}\sum_{k=1}^{n} \alpha \log(\lambda_k) - \log(\alpha\lambda_k + 1 - \alpha)$$

$$= \sum_{k=1}^{n} D_\alpha(\mathcal{N}(0, 1), \mathcal{N}(0, \lambda_k)).$$

Now, we prove that for all $k \in [\![1, n]\!]$, $\lambda_i \in (r_{s,\Delta}, 1/r_{s,\Delta})$. We note $S = v - w$. Any eigenvalue $\lambda$ of $M_{v,w}$ can be written (with a change of variable $x \mapsto \Sigma^{-1/2}x$):

$$\lambda = \frac{x^T\Sigma_w x}{x^T\Sigma_v x},$$

where $x$ is an eigenvector associated to $\lambda$. We have: $\Sigma_w = sI_n + ww^T = sI_n + (v+S)(v+S)^T = \Sigma_v + vS^T + Sv^T + SS^T$.

Then,

$$\lambda - 1 = \frac{x^T(\Sigma_v + vS^T + Sv^T + SS^T)x}{x^T\Sigma_v x} - 1 = \frac{2\langle S^T x, v^T x\rangle + \|S^T x\|^2}{s\|x\|^2 + \|v^T x\|^2}.$$

The condition $\|S\| \leq \Delta$ gives $\|S^T x\|^2 \leq \sigma_{\max}(S)^2\|x\|^2 \leq \Delta^2\|x\|^2$. Cauchy-Schwarz inequality yields $\langle S^T x, v^T x\rangle \leq \|v^T x\|\|S^T x\| \leq \Delta\|v^T x\|\|x\|$, and

$$\lambda \leq 1 + \frac{2\Delta\|v^T x\|\|x\| + \Delta^2\|x\|^2}{s\|x\|^2 + \|v^T x\|^2} \leq \frac{s + \left(\frac{\|v^T x\|}{\|x\|} + \Delta\right)^2}{s + \left(\frac{\|v^T x\|}{\|x\|}\right)^2} \leq \sup_{v \in \mathbb{R}} \frac{s + (v + \Delta)^2}{s + v^2} \leq 1/r_{s,\Delta},$$

using the analysis from Lemma B.5.

Then, leveraging Lemma B.5, we find:

$$\sup_{\|v-w\|\leq\Delta} D_\alpha(\sqrt{s}G + vZ, \sqrt{s}G + wZ) \leq nlD_\alpha(\mathcal{N}(0, s + v_{s,\Delta}^2), \mathcal{N}(0, s + (v_{s,\Delta} + \Delta)^2)),$$

with $v_{s,\Delta} = \frac{1}{2}(\sqrt{\Delta^2 + 4s} - \Delta)$, concluding the proof.

$\square$

Note that there is no reason for this bound to be optimal. In fact, we leverage a upper bound of the eigenvalues of $M_{v,w}$ individually instead of directly analyzing:

$$\sup_{\|v-w\|\leq\Delta} \sum_{k=1}^{n} D_\alpha(\mathcal{N}(0,1), \mathcal{N}(0,\lambda_k)).$$

Performing the same asymptotical analysis as Proposition B.1, we recover the following upper bound for the Rényi divergence:

**Proposition B.2** (Rényi divergence rate of convergence – multivariate case)**.**

$$\sup_{|v-w|\leq\Delta} D_\alpha(\sigma_\theta\sqrt{d - n}G + vZ, \sigma_\theta\sqrt{d - n}G + wZ) \leq \frac{\alpha nl\Delta^2}{4(d - n)\sigma_\theta^2} + o(d^{-1}).$$

### Methodology of experiments — releasing one output.

Let $v, w \in \mathbb{R}^{1\times d}$. We give a result more precise than Theorem 4.3. By Theorem 4.3, there exists $C_{\|v\|,\|w\|} > 0$ such that for all $\alpha \in \left(C_{\|v\|,\|w\|}/d, 1 - C_{\|v\|,\|w\|}/d\right)$:

$$\tilde{G}_{\Lambda(\sigma_\theta,d,\|v\|,\|w\|)}\left(\alpha + \frac{C_{\|v\|,\|w\|}}{d}\right) - \frac{C_{\|v\|,\|w\|}}{d} \leq T(VZ, WZ)(\alpha),$$

Here, the constant $C_{\|v\|,\|w\|}$ is not universal but depends on $v$ and $w$. $C_{\|v\|,\|w\|} = \max\{A_{\|v\|}, A_{\|w\|}\}$ is given by Proposition 4.3. $A_{\|v\|}$ and $A_{\|w\|}$ both depend on a universal constant, derived from Theorem 4.2, that we set to 1 for our experiments. We note $A_{\|v\|} = \left(9 - \frac{6}{(1+d\sigma_\theta^2/\|v\|^2)^2}\right)$.

Based on Lemma B.5, the Rényi divergence $D_\alpha(\mathcal{N}(0, \sigma_\theta d + \|v\|^2), \mathcal{N}(0, \sigma_\theta d + \|w\|^2))$ is maximized by setting $v_{d,\Delta} := \|v\| = \frac{1}{2}(\sqrt{\Delta^2 + 4\sigma_\theta d} - \Delta), w_{d,\Delta} := \|w\| = \frac{1}{2}(\sqrt{\Delta^2 + 4\sigma_\theta d} + \Delta)$. Then, the ratio of the variances $r$ satisfies $r = \frac{\sigma_\theta d + v_{d,\Delta}^2}{\sigma_\theta d + w_{d,\Delta}^2} < 1$. Leveraging Proposition 4.1, for $\alpha \in (0, 1)$, we consider the trade-off function:

$$\tilde{G}_{\Lambda(\sigma_\theta,d,v_{d,\Delta},w_{d,\Delta})}(\alpha) = 2\Phi\left(\frac{\sigma_\theta d + v_{d,\Delta}^2}{\sigma_\theta d + v_{d,\Delta}^2}\Phi^{-1}\left(1 - \alpha/2\right)\right) - 1.$$

Our objective is to estimate the Rényi divergence derived from the following trade-off function:

$$h(\alpha) = \max \begin{cases} T(V, W)(\alpha), \\ 2\Phi\left(\frac{\sigma_\theta d + v_{d,\Delta}^2}{\sigma_\theta d + v_{d,\Delta}^2}\Phi^{-1}\left(1 - \alpha/2 - C_{v_{d,\Delta},w_{d,\Delta}}/2d\right)\right) - 1 + \frac{C_{v_{d,\Delta},w_{d,\Delta}}}{d}, \end{cases}$$

where we set, by abuse of notation:

$$\Phi\left(\frac{\sigma_\theta d + v_{d,\Delta}^2}{\sigma_\theta d + v_{d,\Delta}^2}\Phi^{-1}\left(1 - \alpha/2 - C_{v_{d,\Delta},w_{d,\Delta}}/2d\right)\right) - 1 + \frac{C_{v_{d,\Delta},w_{d,\Delta}}}{d} := 0 \text{ if } \alpha \in (0, C_{v_{d,\Delta},q_{d,\Delta}}/d)\cup(1-C_{v_{d,\Delta},w_{d,\Delta}}/d, 1).$$

The constant $C_{v_{d,\Delta},w_{d,\Delta}}$ is equal to $C_{v_{d,\Delta},w_{d,\Delta}} = \max\{A_{v_{d,\Delta}}, A_{w_{d,\Delta}}\} = \left(9 - \frac{6}{(1+d\sigma_\theta^2/v_{d,\Delta}^2)^2}\right)$.

Now, we can numerically estimate $l_\alpha(h)$.

As discussed in Section 4.3, for $d$ large enough, there exist $0 < c_1 < c_2 < 1$ such that:

$$h(\alpha) = \begin{cases} T(V, W)(\alpha) \text{ if } \alpha \in (0, c_1) \cup (c_2, 1), \\ 2\Phi\left(\frac{\sigma_\theta d + v_{d,\Delta}^2}{\sigma_\theta d + v_{d,\Delta}^2} \Phi^{-1}\left(1 - \alpha/2 - C_{v_{d,\Delta}, w_{d,\Delta}}/2d\right)\right) - 1 + \frac{C_{v_{d,\Delta}, w_{d,\Delta}}}{d} \text{ if } \alpha \in (c_1, c_2). \end{cases}$$

We numerically compute $c_1$ and $c_2$ with the `scipy.optimize.brentq` method, with the smallest possible tolerance `tol = 4*np.finfo(float).eps` $\approx 10^{-15}$, which is negligible compared to the other quantities of the experiment. $\Phi$ and $\Phi^{-1}$ are respectively computed with the methods `scipy.stats.norm.cdf` and `scipy.stats.norm.ppf`. We compute the derivative:

$$h'(\alpha) = \begin{cases} -\frac{\phi(\Phi^{-1}(1-\alpha) - \Delta/\sigma_\theta)}{\phi(\Phi^{-1}(1-\alpha))} \text{ if } \alpha \in (0, c_1) \cup (c_2, 1), \\ -\frac{\sigma_\theta d + v_{d,\Delta}^2}{\sigma_\theta d + v_{d,\Delta}^2} \frac{\phi\left(\frac{\sigma_\theta d + v_{d,\Delta}^2}{\sigma_\theta d + v_{d,\Delta}^2} \Phi^{-1}\left(1 - \alpha/2 - C_{v_{d,\Delta}, w_{d,\Delta}}/2d\right)\right)}{\phi\left(\Phi^{-1}\left(1 - \alpha/2 - C_{v_{d,\Delta}, w_{d,\Delta}}/2d\right)\right)} \text{ if } \alpha \in (c_1, c_2). \end{cases}$$

Then, we estimate $l_\alpha(h)$ using a Monte Carlo algorithm. We set $\sigma_\theta = 1$. We set the number of samples $L = 5 \times 10^5$. We draw $M = 50$ times $(X_{k,k'})_{1 \leq k \leq L, 1 \leq k' \leq M} \overset{iid}{\sim} \text{Unif}([0, 1])$. Then, we run the procedure and the estimates are averaged:

$$\widetilde{l}_\alpha(h) = \frac{1}{M(\alpha - 1)} \sum_{k=1}^M \log \frac{1}{L} \sum_{k'=1}^L |h'(X_{k,k'})|^{1-\alpha}.$$

We repeat the process for different values of $d$ and $\Delta$. For each configuration, we compute the empirical standard variation of the averaged estimates:

$$\tilde{S}_\alpha(h) = \frac{1}{\sqrt{M-1}} \sqrt{\sum_{k=1}^M \left(\frac{1}{\alpha - 1} \log\left(\frac{1}{L} \sum_{k'=1}^L |h'(X_{k,k'})|^{1-\alpha}\right) - \widetilde{l}_\alpha(h)\right)^2}$$

Standard variations are not shown in Figure 2 as they are too small to be distinguishable (they are about 50 times lower than the corresponding averages).

**Methodology of experiments — releasing multiple outputs.**

Let $v, w \in \mathbb{R}^{n \times d}$. By Theorem 4.6, There exists a universal constant $C > 0$ such that for all $\alpha \in \left(Cn\sqrt{\frac{l}{d-n}}, 1 - Cn\sqrt{\frac{l}{d-n}}\right)$:

$$T\left(\sigma_\theta\sqrt{d-n}G + vZ, \sigma_\theta\sqrt{d-n}G + wZ\right)\left(\alpha + Cn\sqrt{\frac{l}{4(d-n)}}\right) - Cn\sqrt{\frac{l}{d-n}} \leq T(VZ, WZ)(\alpha).$$

We take $C = 1$ for our experiments. The left trade-off function is not simple to compute for general values of $v$ and $w$ (see Lemma B.3). However, by independence of rows in $\sigma_\theta\sqrt{d-n}G + vZ$,

$$T\left(\sigma_\theta\sqrt{d-n}G + vZ, \sigma_\theta\sqrt{d-n}G + wZ\right) = T\left(\sigma_\theta\sqrt{d-n}G_1 + vZ_1, \sigma_\theta\sqrt{d-n}G_1 + wZ_1\right)^{\otimes l}.$$

By Lemma B.6 the associated Rényi divergence can be upper bounded by a choice of $v_* = v_{d-n,\Delta}I_{n,d}$, with $v_{d-n,\Delta} = \frac{1}{2}(\sqrt{\Delta^2 + 4\sigma_\theta(d-n)} - \Delta)$, $w_* = (v_{d-n,\Delta} + \Delta)I_{n,d}$. Note that $\|v - w\| = n\Delta$, so the upper bound may not be tight.

We denote $\Phi_{\chi_{nl}^2}$ as the trade-off function of a chi-squared random variable with $nl$ degrees of freedom. Then, we consider

the following trade-off function:

$$T\left(\sigma_\theta\sqrt{d-n}G_1 + v_*Z_1, \sigma_\theta\sqrt{d-n}G_1 + w_*Z_1\right)^{\otimes l}$$

$$=T\left(\sigma_\theta\sqrt{d-n}G_1 + v_{d-n,\Delta}I_{n,d}Z_1, \sigma_\theta\sqrt{d-n}G_1 + (v_{d-n,\Delta}+\Delta)I_{n,d}Z_1\right)^{\otimes l}$$

$$=T\left(\sigma_\theta\sqrt{d-n}G_{1,1} + v_{d-n,\Delta}Z_{1,1}, \sigma_\theta\sqrt{d-n}G_{1,1} + (v_{d-n,\Delta}+\Delta)Z_{1,1}\right)^{\otimes nl}$$

$$=\Phi_{\chi_{nl}^2}\left(\frac{\sigma_\theta^2(d-n) + v_{d-n,\Delta}^2}{\sigma_\theta^2(d-n) + (v_{d-n,\Delta}+\Delta)^2}\Phi_{\chi_{nl}^2}^{-1}(1-\alpha)\right),$$

which is obtained from Lemma B.3. Let the trade-off function:

$$h(\alpha) = \max\begin{cases} T(V,W)(\alpha), \\ \Phi_{\chi_{nl}^2}\left(\frac{\sigma_\theta^2(d-n)+v_{d-n,\Delta}^2}{\sigma_\theta^2(d-n)+(v_{d-n,\Delta}+\Delta)^2}\Phi_{\chi_{nl}^2}^{-1}\left(1-\alpha-Cn\sqrt{\frac{l}{d-n}}\right)\right) - Cn\sqrt{\frac{l}{d-n}}. \end{cases}$$

For $d$ large enough, there exist $0 < c_1 < c_2 < 1$ such that:

$$h(\alpha) = \begin{cases} T(V,W)(\alpha) \text{ if } \alpha \in (0,c_1) \cup (c_2,1), \\ \Phi_{\chi_{nl}^2}\left(\frac{\sigma_\theta^2(d-n)+v_{d-n,\Delta}^2}{\sigma_\theta^2(d-n)+(v_{d-n,\Delta}+\Delta)^2}\Phi_{\chi_{nl}^2}^{-1}\left(1-\alpha-Cn\sqrt{\frac{l}{d-n}}\right)\right) - Cn\sqrt{\frac{l}{d-n}} \text{ if } \alpha \in (c_1,c_2). \end{cases}$$

With the same method as the one output case, we numerically compute $c_1$ and $c_2$. Then, we compute the derivative:

$$h'(\alpha) = \begin{cases} -\frac{\phi(\Phi^{-1}(1-\alpha)-\Delta/\sigma_\theta)}{\phi(\Phi^{-1}(1-\alpha))} \text{ if } \alpha \in (0,c_1) \cup (c_2,1), \\ -\frac{\sigma_\theta^2(d-n)+v_{d-n,\Delta}^2}{\sigma_\theta^2(d-n)+(v_{d-n,\Delta}+\Delta)^2} \frac{\phi_{\chi_{nl}^2}\left(\frac{\sigma_\theta^2(d-n)+v_{d-n,\Delta}^2}{\sigma_\theta^2(d-n)+(v_{d-n,\Delta}+\Delta)^2}\Phi_{\chi_{nl}^2}^{-1}\left(1-\alpha-Cn\sqrt{\frac{l}{d-n}}\right)\right)}{\phi_{\chi_{nl}^2}\left(\Phi_{\chi_{nl}^2}^{-1}\left(1-\alpha-Cn\sqrt{\frac{l}{d-n}}\right)\right)} \text{ if } \alpha \in (c_1,c_2), \end{cases}$$

where $\phi_{\chi_{nl}^2}$ is the density of a chi-squared random variable with $nl$ degrees of freedom. The functions $\Phi_{\chi_{nl}^2}$, $\Phi_{\chi_{nl}^2}^{-1}$ and $\phi_{\chi_{nl}^2}$ are respectively computed with the methods `scipy.stats.chi2.cdf`, `scipy.stats.chi2.ppf` and `scipy.stats.chi2.cdf`. Using the same Monte Carlo procedure as above, we can now compute the averaged empirical estimates with $M = 50$ repetitions and $L = 5 \times 10^5$ samples. We set $l = 10$ and $n = 1$, and we report in Figure 3 the estimated Rényi divergence for multiple values of $d$ and $\Delta$. As in the one-dimensional output case, standard variations are not shown in Figure 3 as they are too small to be distinguishable (they are about 20 times lower than the corresponding averages).

