# OpenReview forum: "Privacy Amplification Through Synthetic Data: Insights from Linear Regression"
_ICML.cc/2025/Conference — ICML 2025 poster_

### Official Review · Reviewer_pnmm · 2025-02-15

**Overall Recommendation:** 3

**Summary:**

The paper offers a theoretical analysis of the privacy loss of releasing synthetic samples in linear regression. It demonstrates that, under a strong threat model where an adversary controls the seed of the generative model, releasing even a single synthetic sample can result in privacy leakage equivalent to that of releasing the full generative model in the worst case. Conversely, when the seed is random, the authors prove a form of privacy amplification.

**Claims And Evidence:**

There are a couple of points that I find problematic or unclear:
- "It is clear that the adversary can recover the model parameters $v^*$ from $d$ queries...Strikingly, we now show that the
adversary can in fact recover the model parameter with just one query." --> The discussion here is very confusing. Privacy leakage is not about recovering model parameters (as a matter of fact, they are already known after training), but rather about inferring information about the training samples. Additionally, what you really show here is that with one query it is possible to achieve the maximum privacy leakage (as specified by the privacy budget of training the generative model) for some worst-case datasets.
- "Since Label DP is a weaker notion than standard DP, these results also imply negative results for standard DP" --> I don't follow this claim. How does a construction where releasing a single synthetic sample achieves maximum privacy leakage under Label DP translate into a construction for standard DP?
- "However, these results do not imply that for every possible seed $z$, the privacy loss $T(V_\infty z, W_\infty z)$ is strictly smaller than $T(V_\infty, W_\infty)$" --> I'm confused about this argument. Mathematically, it seems to be the case that $\|A\Sigma X^\top (y_i-y_i')\| \le \|A X^\top (y_i-y_i')\|$ does not necessarily hold. On the other hand, the post-processing property of DP guarantees that the privacy loss of releasing a single data point is upper bounded by the privacy loss of training the generative model. How can these two observations be reconciled?

**Essential References Not Discussed:**

For DP synthetic data generation, the authors should discuss [1]. In particular, Figure 1 offers a useful overview of the current state of the field.

At a high level, the phenomenon uncovered in this work resembles privacy amplification by iteration: releasing only the final model checkpoint, rather than all the intermediate ones, leads to better privacy. In addition to Feldman et al., 2018, the authors should consider discussing [2,3,4], which provide last-iterate privacy analysis under the assumptions that the loss function is convex and/or smooth, showing that the privacy loss is bounded as $T$ goes to infinity. Moreover, it would be beneficial to review several works on privacy amplification by subsampling [5,6]. Collectively, these studies highlight the power of randomness in privacy protection.

[1] Hu, Yuzheng, et al. "Sok: Privacy-preserving data synthesis." 2024 IEEE Symposium on Security and Privacy (SP). IEEE, 2024.

[2] Altschuler, Jason, and Kunal Talwar. "Privacy of noisy stochastic gradient descent: More iterations without more privacy loss." Advances in Neural Information Processing Systems 35 (2022): 3788-3800.

[3] Ye, Jiayuan, and Reza Shokri. "Differentially private learning needs hidden state (or much faster convergence)." Advances in Neural Information Processing Systems 35 (2022): 703-715.

[4] Chien, Eli, and Pan Li. "Convergent privacy loss of noisy-sgd without convexity and smoothness." arXiv preprint arXiv:2410.01068 (2024).

[5] Balle, Borja, Gilles Barthe, and Marco Gaboardi. "Privacy amplification by subsampling: Tight analyses via couplings and divergences." Advances in neural information processing systems 31 (2018).

[6] Steinke, Thomas. "Composition of differential privacy & privacy amplification by subsampling." arXiv preprint arXiv:2210.00597 (2022).

**Experimental Designs Or Analyses:**

N/A

**Methods And Evaluation Criteria:**

N/A

**Other Comments Or Suggestions:**

N/A

**Other Strengths And Weaknesses:**

Strengths: The paper provides, to the best of my knowledge, the first theoretical analysis of the privacy loss of the synthetic samples generated by DP-trained generative models. Although the setting appears somewhat toyish, the techniques employed, particularly for releasing multiple points, are non-trivial. Overall, this work could serve as a promising first step toward an important research direction.

Weaknesses: The main factor lowering my overall rating is related to the theoretical claims; I feel that the rigor in this work does not meet the bar for ICML. Additionally, the paper could be strengthened by:
- Including a notation section. For instance, $||\cdot||$ is typically interpreted as the 2-norm, but here it is mostly used as the Frobenius norm. Moreover, $\delta_{ij}$, which appears in both Sec 3.1 and Prop 4.2, is never formally defined.
- Providing an overview of the proof techniques. It would be helpful to discuss the technical challenges and how the paper addresses them.
- Discussing the implications of the main results. For example, how does Theorem 4.8 relate to privacy amplification? What is the relationship between $\tilde{G}_{\Lambda(\sigma_w, d, \Delta)}$ and $T(V,W)$?

**Questions For Authors:**

N/A

**Relation To Broader Scientific Literature:**

The paper contributes to the literature on DP data synthesis by providing a formal analysis of the privacy loss of synthetic samples. It also extends the literature on privacy amplification by identifying an alternative mechanism---privacy amplification through synthetic data---beyond traditional approaches such as subsampling and iteration, once again showcasing the power of randomness in privacy protection.

**Theoretical Claims:**

- For output perturbation, Chaudhuri et al., 2011 assumes certain properties of the loss function, specifically bounded gradient (or equivalently, Lipschitz), to upper bound the L2 sensitivity of the minimizer of the regularized least-square objective. To satisfy this property, they primarily focus on classification losses such as cross-entropy and hinge loss and assume that the samples have bounded norm. In contrast, the current paper directly borrows the results from Chaudhuri et al., 2011 but applies them to linear regression. The assumptions from Chaudhuri et al., 2011 are not formally stated, and the gap between the two settings is not addressed, which is problematic. In fact, the gradient of the square loss is not necessarily bounded without additional assumptions.
- The authors claim that "As a discretized Langevin dynamical system with a convex objective, it is known that $V_t$ converges in distribution to its stationary Gibbs distribution". This claim is made without any references or citations. My understanding is that while Langevin dynamics with a convex objective do converge to the stationary Gibbs distribution in continuous time, this convergence is not guaranteed for discretized processes without further assumptions on the step size. This lack of rigor is concerning.

---

> ### Author Rebuttal · Authors · 2025-04-01
>
> We sincerely thank Reviewer pnmm for providing valuable feedback and pointing out several valid issues. Below, we address and discuss each point.
>
> ## Claims And Evidence
>
> > Privacy leakage is not about recovering model parameters
>
> We agree that our sentence might be confusing. What we mean here is that if the adversary is able to recover the model parameters, then no privacy amplification is possible compared to the privacy guarantee given by post-processing the model (this is what we call "maximum privacy leakage" in this context). We show that a single query is sufficient for an adversary to achieve this maximum privacy leakage (Proposition 3.1).
>
> > How… Label DP translate into a construction for standard DP
>
> DP upper bounds the privacy leakage across all possible pairs of adjacent datasets. In Label DP, adjacent datasets differ only in their labels. Since any two datasets that are adjacent under Label DP remain adjacent under standard DP (where both features and labels can differ), a lower bound on the privacy leakage in the Label DP setting also applies to standard DP.
>
> > Mathematically, it seems to be the case that $|A\Sigma X^T(y_i'- y_i)| \leq |A X^T(y_i'- y_i)|$ does not necessarily hold
>
> You are right, thanks for catching this. There is a minor error in the upper bound. We address this in the "Theoretical claims" section below, where you also raised a related question about the convergence of NGD.
>
> ## Theoretical claims
>
> > Output perturbation is not possible without Lipschitz condition on the objective function
>
> You are right and we thank you for pointing out this oversight. To ensure the loss is Lipschitz, we can assume that $\|x\| \leq M_x, \|y\| \leq M_y$ and limit the parameter space to the centered ball of radius $M_\theta$ (the latter condition is always verified for ridge regression). Such conditions are common in private linear regression analysis, see for e.g. [1]. The objective is then $L$-Lipschitz with $L = M_x^2 M_\theta + M_x M_y + \lambda M_\theta$, allowing us to use the output perturbation mechanism and to keep our results unchanged.
>
>
> [1] Y. X. Wang. Revisiting differentially private linear regression: optimal and adaptive prediction & estimation in unbounded domain. AISTATS 2018.
>
> > Discrete convex Langevin dynamical systems do not necessarily converge
>
> Again, you are right. The ergodicity of the process is required for convergence, and is ensured when the objective function is strongly convex and smooth (see for e.g., [2]), which is the case in our setting. Furthermore, for NGD with full batch training, it can be shown that if the learning rate $\eta$ is sufficiently small, then $V_t$ converges to a normal distribution, which is the Gibbs distribution when $\eta \to 0$. The correct result writes as follows.
>
> >Let $\Sigma = \frac{1}{n}X^T X + \lambda I$, $M = I - 2\eta\Sigma$ and denote by $A$ the square root of $\Sigma^{-1} M$. Without loss of generality, assume that $y$ and $y'$ differ. Assume that $\eta(\lambda + M_x^2/n) <1$. Then:
> $$T(V_\infty,W_\infty) = G_{\sqrt{2}\|A X^T (y-y')\|/n\sigma}.$$
> Moreover, for two given datasets, the adversary can choose $z \in \mathbb{R}^d$ such that:
>     $$T(V_\infty z,W_\infty z) = G_{\sqrt{2}|\sigma_{\max}(A X^T (y-y'))|/n\sigma}.$$
> In particular, if $y'$ and $y$ are adjacent (label DP), the adversary can choose $z \in \mathbb{R}^d$ such that:
>     $$T(V_\infty z,W_\infty z) = T(V_\infty,W_\infty).$$
>
> This result corrects both the confusion about convergence of $V_t$ and Propositions 3.2 to 3.4.
> Due to space limitations, we are not able to give the sketch of proof here, but we are happy to provide it as a follow-up comment to the reviewer.
>
> [2] A. Durmus, S. Majewski, and B. Miasojedow. Analysis of Langevin Monte Carlo via convex optimization. J. Mach. Learn. Res. 20:73,  2019
>
> ## References not discussed
>
> Thank you for highlighting these references. We will add a citation to this recent survey and discuss how our work relates to other privacy amplification results. As you noted, synthetic data release is a distinct phenomenon that extends beyond privacy amplification by iteration. In the latter, the final model is released at the end of private training, while our approach further conceals the model itself and discloses only synthetic data generated from random inputs to the model.

---

> > ### Comment · Reviewer_pnmm · 2025-04-01
> >
> > Thanks for the response. I am raising my score to 3 due to the authors’ efforts in improving the rigor of the paper.

---

> > > ### Author Response · Authors · 2025-04-05
> > >
> > > We thank Reviewer pnmm for raising their score. For completeness, we include a sketch of proof of corrected Propositions 3.2 to 3.4, as outlined in our rebuttal.
> > >
> > > ### Proof of corrected Propositions 3.2 to 3.4
> > >
> > > We consider NGD with the following update: $V_{k+1}^T = V_k^T - \frac{1}{n}\eta \sum_{i=1}^m \nabla_w f(V_k^T,x_i,y_i) + \sqrt{\eta} N_{k+1}$. Note that we changed the scaling of $\eta$ for the noise in order for our results to hold. The gradient is $\sum_{i=1}^m \nabla_w f(V_k^T,x_i,y_i) = X^T(XV_k^T - y) + \lambda V_k^T$. Then, noting $B = \frac{2\eta}{n} X^T y$, we get $V_{k+1}^T= MV_k^T + B + \sqrt{\eta} N_{k+1}$.
> > > Then,
> > > $$V_t^T = M^t V_0^T+ \sum_{k=0}^{t-1} M^{t-1-k} (B + \sqrt{\eta} N_{k+1}),$$
> > > which is composed of independent columns with mean $\mu^i_t = (I-M)^{-1}(I - M^t)B_i$ and covariance $\Sigma^i_t = M^{2m} + \eta \sigma^2 (I - M^2)^{-1}(I-M^{2t})$. Assume that $\eta(\lambda + M_x^2/n) < 1$.
> > >
> > > Then $M^t \to 0$ and:
> > >
> > > $$\mu^i_t \to (I-M)^{-1}B_i = \frac{1}{n}\Sigma^{-1} X^T y_i,$$
> > >
> > > $$\Sigma^i_t \to \eta \sigma^2 (I - M^2)^{-1} = \frac{\sigma^2}{2}(I-2\eta\Sigma)^{-1}\Sigma^{-1} = \frac{\sigma^2}{2}M^{-1}\Sigma^{-1}.$$
> > >
> > > By Levy's continuity theorem, $V_t \to V_\infty \sim N( \Sigma^{-1} X^T y/n, \sigma^2 M^{-1} \Sigma^{-1}/2 \otimes I_n)$ (there is an abuse of notation here because we use the vectorized notation). Note that when $\eta \to 0$, we recover the Gibbs distribution.
> > > We note $B^2 = \Sigma^{-1} M^{-1}$. The square roots are defined because $\Sigma$ and $M$ commute.
> > >
> > > By Lemma A.2, the tradeoff function between $V_\infty$ and $W_\infty$ is $T(V_\infty,W_\infty) = G_{\sum_{i=1}^n ||\mu_i'-\mu_i||_{2M\Sigma/\sigma^2}}$, with $\mu_i' = \frac{1}{n}\Sigma^{-1}X^Ty_i', \mu_i = \frac{1}{n}\Sigma^{-1}X^Ty_i$, and:
> > >
> > > $$||\mu_i'-\mu_i||_{2M\Sigma/\sigma^2}^2 = (y_i'-y_i)^T X \Sigma^{-1} M X^T (y_i'-y_i)/\sigma^2 = 2||AX^T(y_i'-y_i)||^2/n^2\sigma^2.$$
> > >
> > > Furthermore, for $z \in \mathbb{R}^d$, $V_\infty z \sim N((\Sigma^{-1} X^T y_i \cdot z)_i/n, \sigma^2 z^T M^{-1} \Sigma^{-1}z I_n/2)$,
> > >
> > > Then, $$T(V_\infty z,W_\infty z) = G_{\frac{\sqrt{2}||z^T \Sigma^{-1} X^T (y-y')||}{ n\sigma || B z||}}.$$
> > >
> > > By noting the change of variable $u = Bz$ and using the invertibility of $B$, we get:
> > >   $$\sup_{z \neq 0} \frac{||z^T \Sigma^{-1} X^T (y-y')||}{  || B z||} = \sup_{u \neq 0} \frac{||u^T (A X^T (y-y'))||}{ ||u||} = \sigma_{\max}(A X^T (y-y')),$$ which corresponds to the 2-norm of $AX^T(y-y')$ and is obtained by setting $u^*$ the right singular vector corresponding to the largest singular value of $A X^T (y-y')$. In the setting of Label DP, $y'-y$ has rank $1$, so $T(V_\infty z,W_\infty z) = T(V_\infty,W_\infty)$.

---

### Official Review · Reviewer_ukYX · 2025-03-10

**Overall Recommendation:** 3

**Summary:**

This paper investigates privacy amplification from synthetic data release within the specific setting of linear regression.

The authors first establish negative results, showing that an adversary controlling the seed of the generative model can induce the maximum possible privacy leakage from a single query.

Conversely, they demonstrate that generating synthetic data from random inputs amplifies privacy beyond the model's inherent guarantees when releasing a limited number of synthetic data points. The amplification holds in the regime when few synthetic samples are released and the ambient dimension d is large.

This highlights the crucial role of randomization in the privacy of synthetic data generation.

---
## update after rebuttal

The paper presents an interesting theoretical observation, which I appreciate, albeit with limited potential applications. The rebuttal reinforces my stance.

**Claims And Evidence:**

The theoretical claims are supported by proofs.

**Essential References Not Discussed:**

N/A

**Experimental Designs Or Analyses:**

N/A

**Methods And Evaluation Criteria:**

N/A

**Other Comments Or Suggestions:**

It might be helpful to contextualize the amplification results in terms of (eps, delta)-DP, perhaps even in restricted choices of the parameters, to better interpret the quantitative improvement over simple post-processing.

**Other Strengths And Weaknesses:**

Strengths:
- This work provides a decently thorough examination of privacy amplification from synthetic data release in the setting of linear regression, including an impossibility result when the internal randomness of the generator is controlled by an adversary, and a positive result when the internal randomness is actually random.

Weaknesses:
- The amplification only holds when the number of generated points is less than the dimension. However, we need $O(d)$ samples to learn the parameters of a linear model, so it doesn't seem possible to learn the linear model using synthetic data while satisfying privacy amplification.
- Another potential weakness is the limitations of the model as well as techniques, which are unlikely to extend beyond linear regression to, e.g. neural networks, where synthetic data is much more useful.

That being said, I believe the theoretical results are interesting within their scope and can be a plausible addition to ICML.

**Questions For Authors:**

1. Are there any downstream applications the authors envision for releasing $< d$ private synthetic data points from a linear model?

**Relation To Broader Scientific Literature:**

The key contribution is the formal proof that releasing synthetic data can in fact reduce privacy loss compared to releasing a privatized model. This is in contrast to most prior works where the privacy loss is bounded using post-processing once there is a privately learned generative model. While privacy amplification from synthetic data is not completely new, past works only study the simpler case of univariate Gaussian data (Neunhoeffer et al., 2024).

**Theoretical Claims:**

I briefly checked the proofs but not in thorough detail.

---

> ### Author Rebuttal · Authors · 2025-04-01
>
> We thank Reviewer ukYX for their feedback. Below, we address each concern separately.
>
> ## Weaknesses
>
> > The amplification only holds when the number of generated points is less than the dimension
>
> This is correct. However, our results do not imply that privacy amplification does not happen when the number of generated points is larger. Our work focuses on the theoretical existence of privacy amplification by synthetic data release. Making it more generally applicable is an interesting open problem.
>
> > "... limitations of the model as well as techniques, which are unlikely to extend beyond linear regression"
>
> We agree that the generalization of our results to deeper models present significant challenges. However, we believe our findings have the potential to be leveraged in broader settings. For instance, the post-processing theorem ensures that the results also apply to regression problems with activation functions---such as logistic or ReLU regression---provided that Lipschitzness and convexity are preserved.
>
> A promising direction for broader applicability is private fine-tuning of the last layer in deeper neural networks, which maintains the linear regression framework. However, modeling the distribution of the noise in this setting becomes more challenging, as the transformation of the Gaussian input through the layers alters its statistical properties. We leave this exploration for future work.
>
> ## Questions
>
> > "Are there any downstream applications the authors envision for releasing $\leq d$ private synthetic data points from a linear model?"
>
>
> As you pointed out in your comment, revealing less than $d$ synthetic data points has limited utility. However, our work is the first to highlight scenarios where amplification is provable, laying the groundwork for deeper theoretical exploration in broader, more realistic contexts. We are confident that our privacy bounds can be extended in practical settings. This is an objective for future work.

---

### Official Review · Reviewer_c2xJ · 2025-03-14

**Overall Recommendation:** 4

**Summary:**

This paper explored the privacy amplification properties of hiding the generative model in private synthetic data generative contexts. Differentially private generative models produce synthetic data that formally inherits the same privacy guarantees. In practice, it has been observed that when the synthetic data generated is small enough, it meets stronger guarantees than the generating model, through an amplification effect. This paper formally shows that this amplification effect exists in cases where synthetic data is generated from random inputs to private linear regression models as case study. In particular, releasing synthetic data leads to stronger privacy guarantees than releasing the generative models when the number of released samples is small enough. The paper also demonstrates that in the case where the adversary has access to the seed of the generative algorithm, there is no such amplification of privacy.

**Claims And Evidence:**

The claims in the paper are well supported by theorems, propositions and lemmas. I reviewed the theoretical results in the main paper, which appear well structured and correct. I did not review proofs and other results in the Appendix.

**Essential References Not Discussed:**

I don't think any essential related work was left out of the discussion.

**Experimental Designs Or Analyses:**

N/A

**Methods And Evaluation Criteria:**

The goal of the paper is to provide an initial theoretical framework to study the phenomenon of privacy amplification through synthetic data. This is achieved mainly via theoretical analysis that is appropriate with respect to the overall goal.

**Other Comments Or Suggestions:**

No additional comments or suggestions at the moment.

**Other Strengths And Weaknesses:**

Strengths:
- The paper addresses an important open question by developing a theoretical framework for quantifying privacy guarantees in synthetic data release, specifically in the context of linear regression. This rigorous approach helps fill a gap in understanding privacy amplification in generative models.
- The paper presents both positive and negative results. It demonstrates that privacy amplification is possible under certain conditions (with random inputs) while also highlighting scenarios where the privacy benefits don't hold, such as when an adversary controls the synthetic data generation seed.

Limitations:
- Restricting the focus to linear regression provides a clean case study but limits the generalizability of the findings: it’s unclear how well these results could extend to more complex models.
- As stated by the author(s), while these findings lay the ground for better insights into private synthetic data, their practical impact is limited.

**Questions For Authors:**

No specific questions at the moment.

**Relation To Broader Scientific Literature:**

The main contribution of the paper is to set up the theoretical framework to study privacy amplification via synthetic data, a phenomenon that was empirically highlighted in work by Annamalai et al. (2024), and in part explored by Neunhoeffer (2024) in a more limited context where training data is one-dimensional and the generative model is a Gaussian with mean and variance estimated privately from the data. This paper uses linear regression to study the phenomenon in a more extended way. The contribution is two-fold: i) the author(s) first prove a negative result: for both output perturbation and noisy gradient descent as methods to privately train the generative model, releasing synthetic data from fixed inputs does not lead to privacy amplification (Theorem 3.1 and 3.4 respectively); ii) then, the paper proves privacy amplification for the single release case (Theorem 4.8) and the more general case of multiple releases (Theorem 4.11). To the best of my knowledge, these contributions are novel, and pave the way for new valuable results in this line of research.

**Theoretical Claims:**

I checked all proofs and results in the main text, and to the best of my knowledge they seem correct. I did not however have the time to review results in Appendix.

---

> ### Author Rebuttal · Authors · 2025-04-01
>
> We thank Reviewer c2xJ for their interesting and positive feedback. Below, we address each concern separately.
>
> ## Limitations
>
> > Restricting the focus to linear regression provides a clean case study but limits the generalizability of the findings: it’s unclear how well these results could extend to more complex models.
>
> We agree that the generalization of our results to deeper models present significant challenges. However, we believe our findings have the potential to be leveraged in broader settings. For instance, the post-processing theorem ensures that the results also apply to regression problems with activation functions---such as logistic or ReLU regression---provided that Lipschitzness and convexity are preserved.
>
> A promising direction for broader applicability is private fine-tuning of the last layer in deeper neural networks, which maintains the linear regression framework. However, modeling the distribution of the noise in this setting becomes more challenging, as the transformation of the Gaussian input through the layers alters its statistical properties. We leave this for future work.

---

### Official Review · Reviewer_RSvt · 2025-03-18

**Overall Recommendation:** 2

**Summary:**

This paper investigates the privacy amplification effect that could be gained when hiding the model that has been used to generate differentially-private synthetic data. The objective is to be able to quantify the privacy gain obtained by releasing only a limited number of synthetic data and not the model itself. More precisely, the authors show that releasing a number of synthetic profiles smaller than the input dimension provides strong privacy guarantees.

**Claims And Evidence:**

Currently, the paper does not contain any experiments for validating the theoretical claims made. If possible, it would have been great to conduct some auditing experiments on controlled datasets to be able to verify these claims.

**Essential References Not Discussed:**

I do not see any missing important references, rather the authors have done a good job at reviewing the corresponding state-of-the-art.

**Experimental Designs Or Analyses:**

The theoretical analysis seem sound although as mentioned earlier I do not have the technical expertise to validate them fully. However, there is no experiments set up for validating them.

**Methods And Evaluation Criteria:**

The paper takes a novel approach of trying to model the worst-case approach for the generative process by giving the control of the seed to the adversary. While this approach is promising, there is however no experimental methodology proposed for validating the performance of such adversarial approach in practice.

**Other Comments Or Suggestions:**

A small typo : « Without loss of generarilt » -> « Without loss of generality »

**Other Strengths And Weaknesses:**

The paper is well-written and the authors have done a good job at explaining the current state-of-the-art on the evaluation of DP guarantees of synthetic data. The theoretical analysis conducted is interesting but only holds for a very small release of synthetic data and thus I consider that the term "privacy amplification" used in the title is exaggerated. There is also a lack of experiments to validate practically the theoretical claims.

**Questions For Authors:**

It would be great if the authors could comment on the potential of the approach to generalize to other types of models.

**Relation To Broader Scientific Literature:**

The main results of the paper contributes to better understand the privacy guarantees that are possible in a context in which only synthetic data is released and not the model itself. However, the impact is limited in the sense that the results only holds if the number of released synthetic samples is very small compared to the input dimension, which is very limited in terms of practical interests.

**Theoretical Claims:**

The theoretical claims are made with respect to two different variants of differential privacy, namely f-DP and Rényi DP. Ideally, it would have been great if the authors could have elaborated on why such notions are necessary compared to the classical DP definition.

Nonetheless, the authors have been able to show that in the specific case of differentially-private linear regression, there exists situation in which if the adversary is able to manipulate the randomness used by the generative process, he can achieve the theoretical upper bound in terms of privacy leakage.

To be frank, the proofs are highly technical and specialized and I do not have the expertise to validate them thoroughly.

---

> ### Author Rebuttal · Authors · 2025-04-01
>
> We thank Reviewer RSvt for their feedback. Below, we address each concern separately.
>
> ## Weaknesses
>
> > The term "privacy amplification" used in the title is exaggerated
>
> In our paper, the phrase "privacy amplification from synthetic data release" refers to potential privacy gains achieved by releasing only synthetic data while keeping the generative model hidden. We demonstrate that this privacy amplification does not occur when the adversary controls the seed. However, existing empirical studies suggest the existence of this effect when the seed is randomized (e.g., [1]). This empirical observation motivates our research question: *Can privacy amplification occur from synthetic data release, or are existing membership inference attacks simply insufficiently powerful to achieve the maximal privacy leakage?*
>
> To address this, we conduct a rigorous theoretical analysis in a simplified linear regression setting. Our results are the first to show that under certain conditions, privacy amplification can indeed occur—even achieving perfect privacy as $d$ increases. While our analysis applies to a specific setting, it does not rule out amplification in more general cases. Instead, our work highlights scenarios where amplification is provable, laying the groundwork for deeper theoretical exploration in broader, more realistic contexts. This motivation is reflected in our title: "Insights from linear regression".
>
> [1] Annamalai, M. S. M. S., Ganev, G., and Cristofaro, E. D. "What do you want from theory alone?" Experimenting with tight auditing of differentially private synthetic data generation. USENIX Security 2024
>
> > There is also a lack of experiments to validate practically the theoretical claims
>
> This is a theoretical paper, and experiments are not necessary to support rigorously proven claims. However, we agree that empirically estimating the privacy guarantees, for example to assess the tightness of our theoretical results, is an interesting idea and we thank the reviewer for this suggestion.
>
> ## Questions
>
> > It would be great if the authors could comment on the potential of the approach to generalize to other types of models.
>
> We agree that the generalization of our results to deeper models present significant challenges. However, we believe our findings have the potential to be leveraged in broader settings. For instance, the post-processing theorem ensures that the results also apply to regression problems with activation functions---such as logistic or ReLU regression---provided that Lipschitzness and convexity are preserved.
>
> A promising direction for broader applicability is private fine-tuning of the last layer in deeper neural networks, which maintains the linear regression framework. However, modeling the distribution of the noise in this setting becomes more challenging, as the transformation of the Gaussian input through the layers alters its statistical properties. We leave this for future work.
>
>
>
> > About $f$-DP and Rényi DP: "Ideally, it would have been great if the authors could have elaborated on why such notions are necessary compared to the classical DP definition."
>
> We chose to consider $f$-DP because it is a tight way to track the privacy guarantees at all $(\epsilon,\delta(\epsilon))$ budgets as trade-off functions. In fact, $f$-DP is the most informative DP notion for the Blackwell order [2]. While alternatives such as privacy profiles could also be considered, our analysis fundamentally relies on the approximation of trade-off functions. For other privacy definitions, this would require other tools and may lead to looser results. In addition to $f$-DP, we considered Rényi DP for its more interpretable privacy bounds, which are easier to grasp than trade-off functions.
>
> [2] Jinshuo Dong, Aaron Roth, and Weijie J Su. Gaussian differential privacy. Journal of the Royal Statistical Society Series B: Statistical Methodology, 84(1):3–37, 2022.

---

### Decision · Program_Chairs · 2025-05-01

**Decision:**

Accept (poster)

**Comment:**

This paper studies the problem of private synthetic data generation in linear regression.

The reviewers agree that the paper presents interesting theoretical results and novel analytic tools. The authors also try their best to address the reviewers' concerns, especially the one regarding the technical flaws. Since most of the reviewers support the paper, I recommend weak accept.